# Spherical Frustum Sparse Convolution Network for LiDAR Point Cloud Semantic Segmentation

**Yu Zheng[1]***, **Guangming Wang[2]***, **Jiuming Liu[1]**, **Marc Pollefeys[3]**, **Hesheng Wang[1]†**

[1] Department of Automation, Shanghai Jiao Tong University
[2] University of Cambridge [3] ETH Zürich

{zhengyu0730,liujiuming,wanghesheng}@sjtu.edu.cn
gw462@cam.ac.uk   marc.pollefeys@inf.ethz.ch

## Abstract

LiDAR point cloud semantic segmentation enables the robots to obtain fine-grained semantic information of the surrounding environment. Recently, many works project the point cloud onto the 2D image and adopt the 2D Convolutional Neural Networks (CNNs) or vision transformer for LiDAR point cloud semantic segmentation. However, since more than one point can be projected onto the same 2D position but only one point can be preserved, the previous 2D projection-based segmentation methods suffer from inevitable quantized information loss, which results in incomplete geometric structure, especially for small objects. To avoid quantized information loss, in this paper, we propose a novel spherical frustum structure, which preserves all points projected onto the same 2D position. Additionally, a hash-based representation is proposed for memory-efficient spherical frustum storage. Based on the spherical frustum structure, the Spherical Frustum sparse Convolution (SFC) and Frustum Farthest Point Sampling (F2PS) are proposed to convolve and sample the points stored in spherical frustums respectively. Finally, we present the Spherical Frustum sparse Convolution Network (SFCNet) to adopt 2D CNNs for LiDAR point cloud semantic segmentation without quantized information loss. Extensive experiments on the SemanticKITTI and nuScenes datasets demonstrate that our SFCNet outperforms previous 2D projection-based semantic segmentation methods based on conventional spherical projection and shows better performance on small object segmentation by preserving complete geometric structure. Codes will be available at https://github.com/IRMVLab/SFCNet.

## 1   Introduction

Nowadays, 3D LiDAR point clouds are widely used sensor data in autonomous robot systems. Many recent works focus on resolving perception [1, 2] and localization [3, 4, 5] tasks on autonomous robot systems using LiDAR point clouds. Among them, semantic segmentation on the LiDAR point cloud enables the robot a fine-grained understanding of the surrounding environment. In addition, the semantic segmentation results can be adopted for the reconstruction of the semantic map [6, 7, 8, 9, 10] of the environments.

Inspired by the achievements of deep learning in image semantic segmentation, researchers focus on searching for effective approaches to transfer the achievements to the field of point cloud semantic segmentation. Most previous works convert the raw point cloud to regular grids, like 2D images [11, 12, 13, 14, 15, 16, 17, 18, 19, 20, 21] and 3D voxels [22, 23, 24, 25], to exploit Convolutional Neural

---

*Equal Contribution.
†Corresponding Author.

38th Conference on Neural Information Processing Systems (NeurIPS 2024).

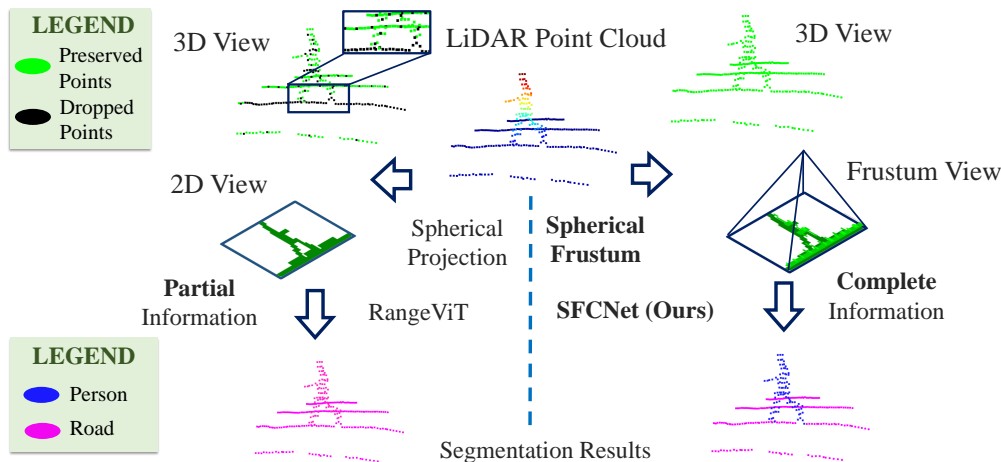

Figure 1: **Difference between our spherical frustum and conventional spherical projection.** In conventional spherical projection, the points projected onto the same 2D grid are dropped, which leads to quantized information loss, *e.g.*, dropping the boundary between the person, a small object, and the road, and results in incorrect prediction of the 2D projection-based method RangeViT [21] for the person. In contrast, our spherical frustum preserves all points in the frustum, which eliminates quantized information loss and makes SFCNet correctly segment the person.

Networks (CNNs) and transformers in the field of point cloud semantic segmentation. The CNNs and transformer can easily process the regular grids to effectively segment the point cloud. However, *due to the limited resolution, more than one point can be projected onto the same grid, and only one point is preserved*, which results in quantized information loss to the regular grid-based point cloud semantic segmentation methods. The quantized information loss poses a challenge for small object segmentation since most points belonging to the small objects can be dropped during the projection. A few methods [13, 17] are proposed to compensate for the quantized information loss by restoring complete semantic predictions from partial predictions. However, quantized information loss still exists in the feature aggregation.

To overcome quantized information loss during 2D projection, in this paper, a novel spherical frustum structure is proposed. Fig. 1 shows the comparison between the conventional spherical projection [11] and spherical frustum. Through spherical frustum, all the points projected onto the same 2D grid are preserved. Therefore, spherical frustum can avoid quantized information loss during the projection and improve the segmentation of small objects. However, without specific designs, the spherical frustum is an irregular structure and can not be processed by CNNs. Using dense grids to store the spherical frustums is an intuitive method to regularize the spherical frustum. However, since the point number of the spherical frustums is different, each point set is required to be padded to the maximal point number of the spherical frustums before being stored in the dense grid, which results in many redundant memory costs. To avoid redundant memory occupancy, we propose a hash-based spherical frustum representation, which stores spherical frustums in a memory-efficient way. In the hash-based spherical frustum representation, the neighbor relationship of spherical frustums and points is represented through the hash table, which enables the points to be simply stored in the original irregular point set.

In the hash-based representation, each point is uniquely identified by the hash key, which consists of the 2D coordinates of the corresponding spherical frustum and the point index in the spherical frustum point set. Thus, the points projected onto any specific 2D grids can be efficiently queried. Based on the hash-based representation, we propose the Spherical Frustum sparse Convolution (SFC) to exploit 2D CNNs on spherical frustums. SFC aggregates point features of nearby spherical frustums to obtain the local feature of the center point.

Moreover, previous 2D projection-based segmentation methods downsample the projected point cloud based on stride-based 2D sampling, which is unable to uniformly sample the 3D point cloud. However, the stride-based 2D sampling uniformly samples the spherical frustums. Therefore, we propose a novel uniform point cloud sampling method, Frustum Farthest Point Sampling (F2PS).

F2PS firstly samples spherical frustums by stride, and then uniformly samples the point set inside each sampled spherical frustum by Farthest Point Sampling (FPS) [26]. Since the computing complexity of sampling points in each spherical frustum is constant-level, F2PS is an efficient sampling algorithm with a linear computing complexity.

In summary, our contributions are:

- We propose a novel spherical frustum structure with a memory-efficient hash-based representation. Spherical frustum avoids quantized information loss of spherical projection and preserves complete geometric structure.

- We integrate spherical frustum structure into 2D sparse convolution, and propose a novel Spherical Frustum sparse Convolution Network (SFCNet) for LiDAR point cloud semantic segmentation.

- An efficient and uniform 3D point cloud sampling named Frustum Farthest Point Sampling (F2PS) is proposed based on the spherical frustum structure.

- SFCNet is evaluated on the SemanticKITTI [1] and nuScenes [27] datasets. The experiment results show that SFCNet outperforms previous 2D projection-based methods and can better segment small objects.

## 2   Related Work

**Point-Based Semantic Segmentation.**   A group of works [26, 28, 29, 30, 31, 32, 33] learn to segment point cloud based on the raw unstructured point cloud. However, learning of raw point cloud requires the neighborhood query with high computing complexity to learn effective features from the local point cloud structure. Therefore, the efficiency of these point-based methods is limited.

**3D Sparse Voxel-Based Semantic Segmentation.**   Storing large-scale LiDAR point clouds in dense 3D voxels requires huge memory consumption. Therefore, Graham et al [34] proposes the 3D sparse voxel structure. Instead of dense grids, the hash table is adopted to represent the neighborhood relations of the 3D sparse grids. Based on the hash table, the convolved grids are recorded in the rule book. According to the rule book, the 3D sparse convolution is performed. Based on the sparse 3D voxel architecture, the methods of 3D sparse convolution and 3D attention mechanisms [22, 23, 24, 35, 36, 25, 37] are proposed.

**2D Projection-Based Semantic Segmentation.**   The research of image semantic segmentation [38, 39, 40, 41, 42] has gained great achievement. Thus, many works [11, 12, 13, 14, 15, 43, 18, 19, 20, 21, 16, 17] project the point cloud onto the 2D plane and utilize 2D neural networks to process the projected point cloud. Spherical projection is a widely used projection method first introduced by SqueezeSeg [11]. The subsequent works [11, 12, 13, 14, 43, 20, 21] effectively segment the point cloud with the image semantic segmentation architecture including 2D CNNs and vision transformers.

Due to the limited resolution, the 2D projection-based segmentation methods suffer from quantized information loss. With quantized information loss, networks can only process the incomplete geometric structure and output partial semantic predictions, which results in the penalty of segmentation performance. The previous works only focus on restoring complete semantic predictions from the partial predictions of 2D neural networks. RangeNet++ [13] proposes a post-processing strategy to restore the complete predictions. The semantic predictions of dropped points are voted by the predictions of their K-Nearest Neighbors (KNN). In addition to KNN-based post-processing, KPRNet [17] directly reprojects incomplete predictions to the complete point cloud and adopts point-based network KPConv [29] to refine the predictions. However, few works explore the method of preserving the complete geometric structure during projection.

In this paper, we propose the spherical frustum which avoids the quantized information loss of spherical projection. Our spherical frustum structure can not only preserve the complete geometric structure but also output the complete semantic predictions without any post-processing or point-based network refinement.

# 3 SFCNet

In this section, the spherical frustum and the hash-based representation will be first illustrated in Sec. 3.1. Based on the hash-based spherical frustum representation, the spherical frustum sparse convolution and frustum farthest point sampling for LiDAR point cloud semantic segmentation will be introduced in Sec. 3.2 and 3.3 respectively. Finally, the architecture of the Spherical Frustum sparse Convolution Network (SFCNet) is illustrated in Sec. 3.4.

## 3.1 Spherical Frustum

**Conventional Spherical Projection.** The LiDAR point cloud $\mathcal{P}$ is composed of $N$ points. The $k$-th point in $\mathcal{P}$ is represented by its 3D coordinates $\boldsymbol{x}_k = [x_k, y_k, z_k]^T$ and the input point features $\boldsymbol{f}_k \in \mathbb{R}^{C_{in}}$, where $C_{in}$ represents the channel dimension of the features. The conventional spherical projection [11] first calculates the 2D spherical coordinates of each point:

$$\begin{pmatrix} u_k \\ v_k \end{pmatrix} = \begin{pmatrix} \frac{1}{2}[1 - \arctan(y_k, x_k)\pi^{-1}] \cdot W \\ [1 - (\arcsin(z_k/r_k) + f_{down}) \cdot f^{-1}] \cdot H \end{pmatrix}, \tag{1}$$

where $(H, W)$ is the height and width of the projected image. $r_k = \sqrt{x_k^2 + y_k^2 + z_k^2}$ is the range of the point. $f = f_{up} + f_{down}$ is the vertical field-of-view of the LiDAR sensor, where $f_{up}$ and $f_{down}$ are the up and down vertical field-of-views respectively. According to the computed 2D spherical coordinates, the point features $\{\boldsymbol{f}_k\}_{k=1}^N$ are projected onto the 2D dense image. If multiple points have the same 2D coordinates, the conventional spherical projection only projects the point closest to the origin and drops the other points, which results in the quantized information loss.

**From Spherical Projection to Spherical Frustum.** Since dropping the redundant points projected onto the same 2D position results in quantized information loss, we propose the spherical frustum to preserve all the points projected onto the same 2D position. Specifically, we organize these points as a point set and assign each point with the unique index $m_k$ in the point set. In addition, the 3D coordinates of each point $\{(x_k, y_k, z_k)\}_{k=1}^N$ are preserved as the 3D geometric information for the subsequent modules.

**Hash-Based Spherical Frustum Representation.** The irregular spherical frustums can not be directly processed by the 2D CNNs. A natural idea to regularize the spherical frustums is putting them in dense grids. To store the point set of each spherical frustum in the dense grids, an extra grid dimension is required. The size of this dimension should be the maximal point number $M$ of each spherical frustum point set. However, since most of the spherical frustum point numbers are much less than $M$, many grids are empty. To avoid saving these empty grids in memory, we propose the hash-based spherical frustum representation to regularize the spherical frustum, where the hash table replaces the dense grids to map the 2D coordinates to the corresponding spherical frustums and points. In the hash table, the index $k$ of any point in the original point cloud can be queried using the key $(u_k, v_k, m_k)$, which is the combination of the 2D spherical coordinates and the point index in the spherical frustum point set. Based on the hash-based representation, the spherical frustums are regularly stored in a memory-efficient way.

## 3.2 Spherical Frustum Sparse Convolution

Since multiple points are stored in a single spherical frustum, the conventional 2D convolution can not be directly performed on the spherical frustum structure. Therefore, we propose Spherical Frustum sparse Convolution (SFC). As shown in Fig. 2, SFC can be seen as the sparse convolution on the virtual spherical plane of the center point. The feature of each convolved 2D position on the virtual spherical plane is filled with the feature of the nearest point in the corresponding spherical frustum.

**Selecting Convolved Spherical Frustums.** SFC first selects the convolved spherical frustums for each center point $p$. The 3D coordinates and the 2D spherical coordinates of the center point $p$ are $(x, y, z)$ and $(u, v)$ respectively. The conventional 2D convolution convolves the features of the grids in the convolution kernel. Similar to the conventional convolution, the spherical frustum of each 2D position in the convolution kernel is selected to perform the convolution. The coordinates of the 2D positions are $\{(u + \Delta u_i, v + \Delta v_i)\}_{i=1}^{K^2}$, where $K$ is the kernel size and $(\Delta u_i, \Delta v_i)$ are the shift inside the kernel. Then, the points inside each spherical frustum are queried through the hash table. Meanwhile, the features $\{\boldsymbol{f}_j\}_{j=1}^{M_i}$ of these points are obtained, where $M_i$ is the number of the points

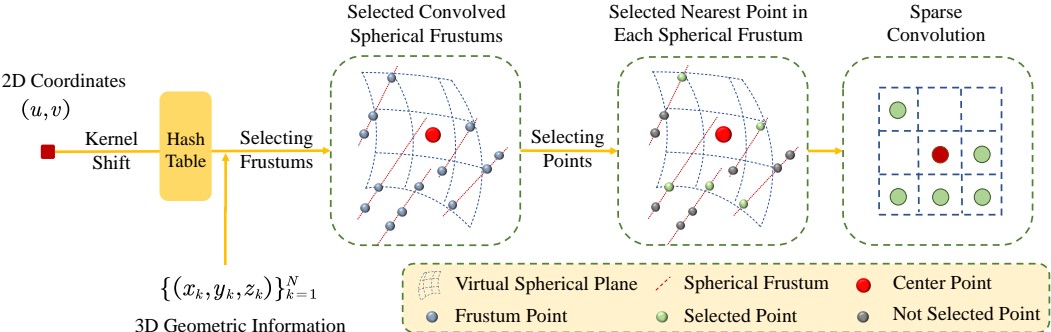

Figure 2: **Pipeline of Spherical Frustum sparse Convolution.** The spherical frustums in the convolution kernel and the points in these spherical frustums are first selected through the hash table. Then, the nearest point in each spherical frustum is determined by the 3D geometric information. Finally, the sparse convolution is performed on the selected point features.

in the $i$-th spherical frustum. Notably, $M_i$ can be zero, which means that no points are projected onto the corresponding 2D position, and the spherical frustum is invalid. The invalid spherical frustums are ignored in the subsequent convolution.

**Selecting the Nearest Point in Each Spherical Frustum.** After identifying the points in the spherical frustum, a feature should be selected from the features of the frustum point set for 2D convolution. PointNet++ [26] emphasizes that the local feature of the center point is expected to be aggregated from the 3D neighboring points. Inspired by PointNet++, we select the feature of the nearest point to the center point in each spherical frustum. Specifically, based on the 3D geometric information, the 3D coordinates $\{(x_j, y_j, z_j)\}_{j=1}^{M_i}$ of the frustum points are obtained for the nearest point selection. Inspired by the post-processing of RangeNet++ [13], we select the distance of range $r_j = \sqrt{x_j^2 + y_j^2 + z_j^2}$ rather than the Euclidean distance as the metric of the nearest point for efficient distance calculation. Therefore, the selected point index of each spherical frustum is $\arg\min_j |r_j - r|$, where $r$ is the range of the center point. According to the indexes, the convolved features $\{\boldsymbol{f}_i\}_{i=1}^{K'}$ are obtained, where $K'$ is the number of valid spherical frustums.

**Sparse Convolution.** Finally, the sparse convolution is performed as:

$$\boldsymbol{f}' = \sum_{i=1}^{K'} W_i \boldsymbol{f}_i, \tag{2}$$

where $W_i$ is the convolution weight of the $i$-th valid 2D position, and $\boldsymbol{f}'$ is the aggregated feature.

Through the proposed spherical frustum sparse convolution, we realize effective regularization and 2D convolution-based feature aggregation for all points in the unstructured point cloud.

### 3.3 Frustum Farthest Point Sampling

Sampling is a significant process of point cloud semantic segmentation. Through sampling, the network can aggregate the features of different scales and recognize objects of different sizes. Moreover, the sampling should uniformly sample the point cloud to avoid key information loss. The previous 2D projection-based methods sample the projected point cloud using stride-based 2D sampling. This sampling ignores the 3D geometric structure of the point cloud. In contrast, as shown in Fig. 3, our Frustum Farthest Point Sampling (F2PS) only samples the spherical frustums by stride, while the spherical frustum point set is sampled by farthest point sampling.

**Sampling Spherical Frustums by Stride.** Specifically, we split the 2D spherical plane into several non-overlapping windows of size $S_h \times S_w$, where $(S_h, S_w)$ are the strides. The spherical frustums in each window are merged as the downsampled spherical frustum. Meanwhile, the points inside the merged spherical frustums are queried through the hash table. Then, the queried points are merged as the point set $\{p_l\}_{l=1}^{L}$ in the downsampled spherical frustum, where $L$ is the point number in the downsampled spherical frustum.

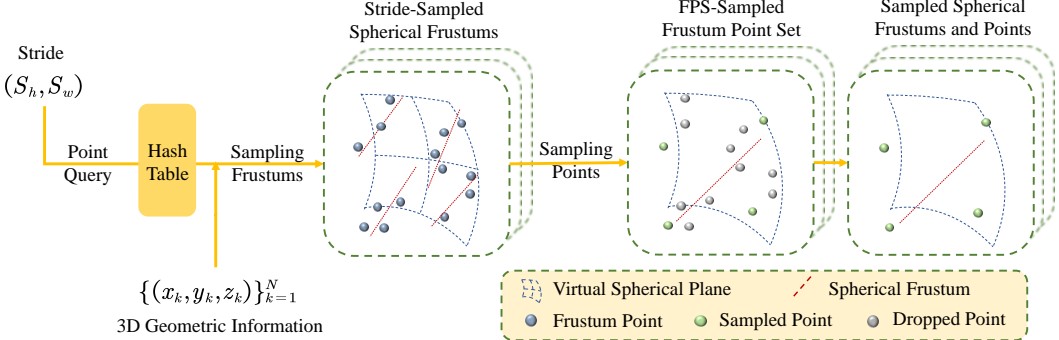

Figure 3: **Pipeline of Frustum Farthest Point Sampling.** According to the downsampling strides, the spherical frustums in each stride window are downsampled. Then, through the hash table, the points in each downsampled spherical frustum are queried. The queried points are sampled by Farthest Point Sampling (FPS) based on the 3D geometric information. Finally, the uniformly sampled spherical frustums and point cloud are obtained.

**Sampling Frustum Point Set by Farthest Point Sampling.** The Farthest Point Sampling (FPS) [26] is adopted to uniformly sample the point set in the downsampled spherical frustum. Since the point number of each downsampled spherical frustum is much smaller than the point number of the point cloud, performing FPS is not time-consuming. Specifically, the 3D coordinates of each point in $\{p_l\}_{l=1}^{L}$ are first acquired from the 3D geometric information for 3D distance calculation. Then, the $\lceil L/(S_h \times S_w) \rceil$ points are iteratively sampled from the original point set. At each iteration, the distance of each non-sampled point towards the sampled point set is calculated. The point that has the maximal distance is added to the sampled set. Finally, the uniformly sampled spherical frustum point set is obtained.

F2PS integrates the stride-based spherical frustum sampling with the FPS-based frustum point set sampling. Thus, F2PS can sample the original point cloud uniformly. In addition, since performing FPS on the frustum point set costs $O(1)$ time, the computing complexity of F2PS is $O(N)$. Thus, F2PS is an efficient point cloud sampling algorithm.

## 3.4 Network Architecture

Based on the Spherical Frustum sparse Convolution (SFC) and Frustum Farthest Point Sampling (F2PS), the Spherical Frustum sparse Convolution Network (SFCNet) is constructed. SFCNet is an encoder-decoder architecture. The hash-based spherical frustum representation is first built for convolution and sampling in the subsequent modules. Then, the point features $\{f \in \mathbb{R}^C\}$ are extracted through the encoder of SFCNet, where $C$ is the channel dimension. The encoder consists of the residual convolutional blocks from ResNet [44], where the convolutions are replaced by the proposed SFC. In addition, the point cloud is downsampled based on F2PS to extract the features of different scales. After each downsampling operation, an SFC layer is adopted to aggregate the neighbor features for each sampled point. Since F2PS can uniformly sample the point cloud, the information of the original point cloud can be fully gathered in the downsampled point cloud. In the decoder, the extracted features are upsampled, concatenated, and fed into the head layer to output the prediction of semantic segmentation.

## 4 Experiments

In this section, we first introduce the datasets adopted in the experiments and the implementation details of the SFCNet. Then, the quantitative results of the two datasets and the qualitative results of the SemanticKITTI dataset are presented. Finally, the ablation studies and comparison with restoring-based methods are conducted to validate the effectiveness of the proposed modules.

Table 1: Quantative results of semantic segmentation on the SemanticKITTI [1] test set. **Bold** results are the best in each block of methods.

| Approach | mIoU (%) | car | bicycle | motorcycle | truck | other-vehicle | person | bicyclist | motorcyclist | road | parking | sidewalk | other-ground | building | fence | vegetation | trunk | terrain | pole | traffic-sign |
|---|---|---|---|---|---|---|---|---|---|---|---|---|---|---|---|---|---|---|---|---|
| **Point Based** | | | | | | | | | | | | | | | | | | | | |
| PointNet++ [26] | 20.1 | 53.7 | 1.9 | 0.2 | 0.9 | 0.2 | 0.9 | 1.0 | 0.0 | 72.0 | 18.7 | 41.8 | 5.6 | 62.3 | 16.9 | 46.5 | 13.8 | 30.0 | 6.0 | 8.9 |
| RandLA [31] | 55.9 | 94.2 | **47.4** | 32.2 | **43.9** | 39.1 | 48.4 | 47.4 | 9.4 | **90.5** | **61.8** | **74.0** | 24.5 | 89.7 | 60.4 | 83.8 | 63.6 | 68.6 | 51.0 | **50.7** |
| KPConv [29] | 58.8 | **96.0** | 30.2 | **42.5** | 33.4 | **44.3** | **61.5** | **61.6** | **11.8** | 88.8 | 61.3 | 72.7 | **31.6** | **90.5** | **64.2** | **84.8** | **69.2** | **69.1** | **56.4** | 47.4 |
| **3D Voxel Based** | | | | | | | | | | | | | | | | | | | | |
| Cylinder3D [23] | 67.8 | 97.1 | 67.6 | 64.0 | 59.0 | 58.6 | 73.9 | 67.9 | 36.0 | 91.4 | 65.1 | 75.5 | 32.3 | 91.0 | 66.5 | 85.4 | 71.8 | 68.5 | 62.6 | 65.6 |
| (AF)²-S3Net [22] | 69.7 | 94.5 | 65.4 | **86.8** | 39.2 | 41.1 | **80.7** | **80.4** | 74.3 | 91.3 | 68.8 | 72.5 | **53.5** | 87.9 | 63.2 | 70.2 | 68.5 | 53.7 | 61.5 | 71.0 |
| SphereFormer [25] | **74.8** | **97.5** | **70.1** | 70.5 | **59.6** | **67.7** | 79.0 | 80.4 | **75.3** | **91.8** | **69.7** | **78.2** | 41.3 | **93.8** | **72.8** | **86.7** | **75.1** | **72.4** | **66.8** | **72.9** |
| **2D Projection Based** | | | | | | | | | | | | | | | | | | | | |
| RangeNet++ [13] | 52.2 | 91.4 | 25.7 | 34.4 | 25.7 | 23.0 | 38.3 | 38.8 | 4.8 | 91.8 | 65.0 | 75.2 | 27.8 | 87.4 | 58.6 | 80.5 | 55.1 | 64.6 | 47.9 | 55.9 |
| PolarNet [16] | 54.3 | 93.8 | 40.3 | 30.1 | 22.9 | 28.5 | 43.2 | 40.2 | 5.6 | 90.8 | 61.7 | 74.4 | 21.7 | 90.0 | 61.3 | 84.0 | 65.5 | 67.8 | 51.8 | 57.5 |
| SqueezeSegV3 [14] | 55.9 | 92.5 | 38.7 | 36.5 | 29.6 | 33.0 | 45.6 | 46.2 | 20.1 | 91.7 | 63.4 | 74.8 | 26.4 | 89.0 | 59.4 | 82.0 | 58.7 | 65.4 | 49.6 | 58.9 |
| SalsaNext [15] | 59.5 | 91.9 | 48.3 | 38.6 | 38.9 | 31.9 | 60.2 | 59.0 | 19.4 | 91.7 | 63.7 | 75.8 | 29.1 | 90.2 | 64.2 | 81.8 | 63.6 | 66.5 | 54.3 | 62.1 |
| KPRNet [17] | 63.1 | **95.5** | 54.1 | 47.9 | 23.6 | 42.6 | 65.9 | 65.0 | 16.5 | **93.2** | **73.9** | **80.6** | 30.2 | 91.7 | 68.4 | **85.7** | 69.8 | **71.2** | 58.7 | 64.1 |
| Lite-HDSeg [19] | 63.8 | 92.3 | 40.0 | 55.4 | 37.7 | 39.6 | 59.2 | 71.6 | **54.1** | 93.0 | 68.2 | 78.3 | 29.3 | 91.5 | 65.0 | 78.2 | 65.8 | 65.1 | 59.5 | **67.7** |
| RangeViT [21] | 64.0 | 95.4 | 55.8 | 43.5 | 29.8 | 42.1 | 63.9 | 58.2 | 38.1 | 93.1 | 70.2 | 80.0 | **32.5** | **92.0** | **69.0** | 85.3 | 70.6 | **71.2** | 60.8 | 64.7 |
| CENet [20] | 64.7 | 91.9 | 58.6 | 50.3 | **40.6** | 42.3 | 68.9 | 65.9 | 43.5 | 90.3 | 60.9 | 75.1 | 31.5 | 91.0 | 66.2 | 84.5 | 69.7 | 70.0 | 61.5 | 67.6 |
| SFCNet (Ours) | **65.0** | 95.1 | **64.2** | **63.2** | 23.5 | **45.6** | **78.3** | **73.1** | 26.4 | 87.9 | 65.6 | 71.9 | 29.1 | 91.1 | 64.5 | 83.7 | **72.6** | 69.6 | **62.6** | 67.6 |

Table 2: Quantative results of semantic segmentation on the nuScenes [27] validation set. **Bold** results are the best in each block of methods.

| Approach | mIoU (%) | barrier | bicycle | bus | car | construction | motorcycle | pedestrian | traffic-cone | trailer | truck | driveable | other flat | sidewalk | terrain | manmade | vegetation |
|---|---|---|---|---|---|---|---|---|---|---|---|---|---|---|---|---|---|
| **3D Voxel Based** | | | | | | | | | | | | | | | | | |
| (AF)²-S3Net [22] | 62.2 | 60.3 | 12.6 | 82.3 | 80.0 | 20.1 | 62.0 | 59.0 | 49.0 | 42.2 | 67.4 | 94.2 | 68.0 | 64.1 | 68.6 | 82.9 | 82.4 |
| Cylinder3D [23] | 76.1 | 76.4 | 40.3 | 91.3 | **93.8** | 51.3 | 78.0 | 78.9 | 64.9 | 62.1 | 84.4 | 96.8 | 71.6 | **76.4** | **75.4** | 90.5 | 87.4 |
| SphereFormer [25] | **78.4** | **77.7** | **43.8** | **94.5** | 93.1 | **52.4** | **86.9** | **81.2** | **65.4** | **73.4** | **85.3** | **97.0** | **73.4** | 75.4 | 75.0 | **91.0** | **89.2** |
| **2D Projection Based** | | | | | | | | | | | | | | | | | |
| RangeNet++ [13] | 65.5 | 66.0 | 21.3 | 77.2 | 80.9 | 30.2 | 66.8 | 69.6 | 52.1 | 54.2 | 72.3 | 94.1 | 66.6 | 63.5 | 70.1 | 83.1 | 79.8 |
| PolarNet [16] | 71.0 | 74.7 | 28.2 | 85.3 | 90.9 | 35.1 | 77.5 | 71.3 | 58.8 | 57.4 | 76.1 | 96.5 | 71.1 | **74.7** | 74.0 | 87.3 | 85.7 |
| SalsaNext [15] | 72.2 | 74.8 | 34.1 | 85.9 | 88.4 | 42.2 | 72.4 | 72.2 | 63.1 | 61.3 | 76.5 | 96.0 | 70.8 | 71.2 | 71.5 | 86.7 | 84.4 |
| RangeViT [21] | 75.2 | 75.5 | **40.7** | 88.3 | 90.1 | **49.3** | 79.3 | 77.2 | **66.3** | 65.2 | 80.0 | 96.4 | 71.4 | 73.8 | 73.8 | **89.9** | 87.2 |
| SFCNet (Ours) | **75.9** | **76.7** | 40.4 | **89.5** | **91.3** | 46.7 | **82.0** | **78.1** | 65.8 | **69.4** | **80.6** | **96.6** | **71.6** | 74.5 | **74.9** | 89.0 | **87.5** |

## 4.1 Datasets

We train and evaluate SFCNet on the SemanticKITTI [1] and nuScenes [27] datasets, which provide point-wise semantic labels for large-scale LiDAR point clouds.

SemanticKITTI [1] dataset contains 43551 LiDAR point cloud scans captured by the 64-line Velodyne-HDLE64 LiDAR. Each scan contains nearly $120K$ points. These scans are split into 22 sequences. According to the official setting, we split sequences 00-07 and 09-10 as the training set, sequence 08 as the validation set, and sequences 11-21 as the test set. Moreover, SemanticKITTI provides the point-wise semantic annotations of 19 classes for the LiDAR semantic segmentation task.

NuScenes [27] dataset consists of 34149 LiDAR point cloud scans collected in 1000 autonomous driving scenes using the 32-line Velodyne HDL-32E LiDAR. Each scan contains nearly $40K$ points. We adopt the official setting to split the scans of the nuScenes dataset into the training and validation

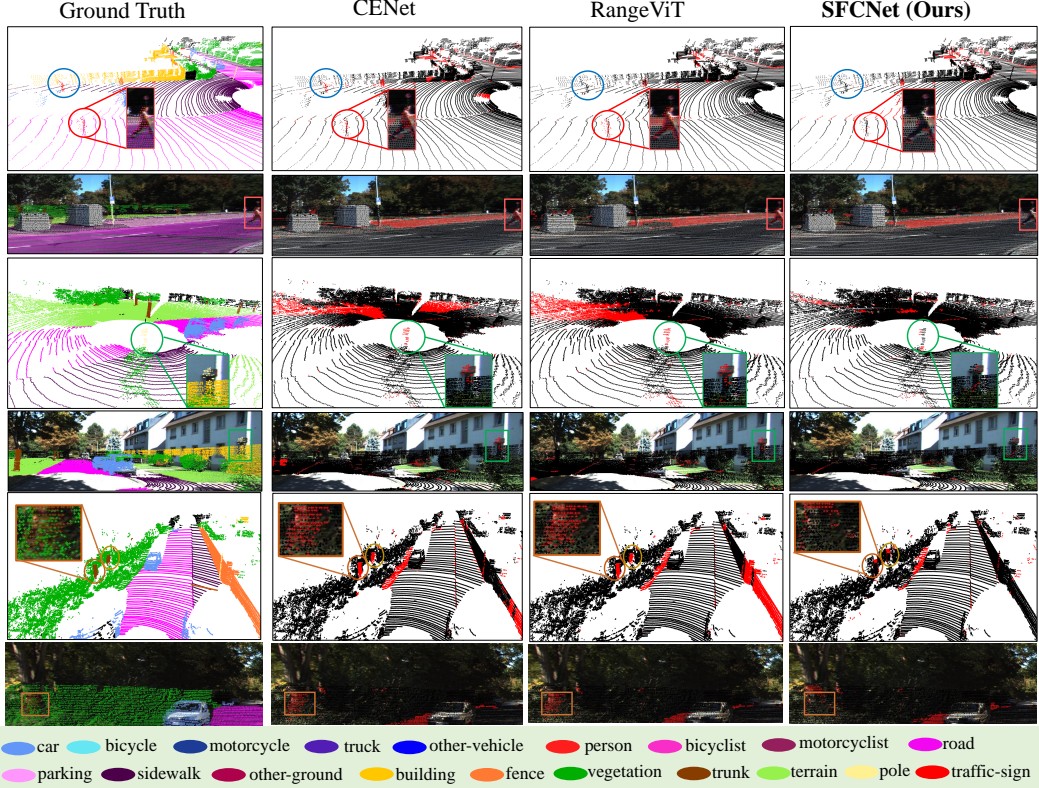

| Ground Truth | CENet | RangeViT | **SFCNet (Ours)** |

car · bicycle · motorcycle · truck · other-vehicle · person · bicyclist · motorcyclist · road
parking · sidewalk · other-ground · building · fence · vegetation · trunk · terrain · pole · traffic-sign

Figure 4: Qualitive results on SemanticKITTI validation set. The first column presents the ground truths, while the following three columns show the error maps of the predictions from the three methods. Specifically, the reference from point color to the semantic class in the ground truths is shown at the bottom. In addition, the false-segmented points are marked as red in the error maps. Moreover, we use circles with the same color to point out the same objects in the ground truth and the three error maps. Furthermore, the corresponding RGB images of each scene with the colored point cloud projected are demonstrated. We also show the corresponding zoomed RGB image view of circled objects if they are visible in the RGB images.

sets. In addition, in nuScenes dataset, 16-class point-wise semantic annotations are provided for the LiDAR semantic segmentation task.

On both datasets, the performance of LiDAR point cloud semantic segmentation is evaluated by mean Intersection-over-Union (mIoU) [45].

### 4.2 Implementation Details

SFCNet is implemented through PyTorch [46] framework. For spherical frustum, the height and width in the calculation of spherical coordinates are set as $H = 64, W = 1800$ for the SemanticKITTI dataset, and $H = 32, W = 1024$ for the nuScenes dataset. The channel dimensions $C$ of the extracted point features for SemanticKITTI and nuScenes datasets are set as $128$ and $256$ respectively. The strides $(S_h, S_w)$ of the F2PS are all set as $(2, 2)$. The multi-layer weighted cross-entropy loss and Lovász-Softmax loss [47] are adopted for network optimization. Adam [48] with the initial learning rate $0.001$ is treated as the optimizer. The learning rate is delayed by $5\%$ in every epoch. Random rotation, flipping, translation, and scaling are utilized for data augmentation on both datasets. Model training is conducted on a single NVIDIA Quadro RTX 8000. The training batch size is set as $4$.

### 4.3 Quantative Results

We compare our SFCNet to the State-of-The-Art (SoTA) 2D projection-based, point-based and 3D voxel-based segmentation methods on the SemanticKITTI and nuScenes datasets.

As shown in Tabs. 1 and 2, SFCNet outperforms the previous SoTA 2D convolution-based segmentation methods CENet [20] and SalsaNext [15] on the SemanticKITTI and nuScenes datasets respectively. In addition, SFCNet also has better performance than the vision transformer-based segmentation method RangeViT [21] on both two datasets. SFCNet also outperforms the point-based methods and realizes a smaller performance gap between the 2D projection-based methods and the 3D voxel-based methods. As for the per-class IoU comparison, SFCNet has better IoU than the other 2D projection-based methods on the small 3D objects, including the motorcycle, person (which is pedestrian in nuScenes), bicyclist, trunk, and pole. The performance improvement on these small objects results from the elimination of quantized information loss. Without quantized information loss, the complete geometric structure of the small 3D objects can be preserved, which enables more accurate segmentation. We also observe the slightly weaker performances on wide-range classes, *e.g.*, road, parking, and terrain, on the SemanticKITTI dataset. However, since preserving complete points significantly improves the accuracies of the hard small objects, SFCNet has a higher mean IoU than the previous 2D projection-based methods.

## 4.4 Qualitative Results

Fig. 4 presents the qualitative comparison between our SFCNet and the 2D projection-based segmentation methods CENet [20] and RangeViT [21]. The comparison shows that the predictions of SFCNet have the minimal segmentation error among the three methods. Moreover, the circled objects in the three rows of Fig. 4 demonstrate the accurate segmentation of SFCNet to the persons, poles, and trunks respectively. This result further indicates our better segmentation performance of 3D small objects by eliminating the quantized information loss.

## 4.5 Ablation Study

In this section, we conduct the ablation study on the SemanticKITTI dataset to validate the effectiveness of the proposed modules. We adopt the baseline network using the conventional spherical projection and stride-based sampling. The results of ablation studies are shown in Tab. 3.

**Spherical Frustum Sparse Convolution (SFC).** First, we replace spherical projection in the baseline with spherical frustum and adopt spherical frustum sparse convolution for feature aggregation. After replacement, the mIoU increases $4.3\%$, which indicates that spherical frustum structure can avoid the quantized information loss, and thus prevent segmentation error from incomplete predictions.

Table 3: Results of ablation studies on the SemanticKITTI validation set.

| ID | Baseline | SFC | F2PS | mIoU (%) |
|---|---|---|---|---|
| 1 | ✓ | | | 56.2 |
| 2 | ✓ | ✓ | | 60.5 |
| 3 | ✓ | ✓ | ✓ | **62.9** |

**Frustum Farthest Point Sampling (F2PS).** After replacing the stride-based 2D sampling with F2PS, the mIoU increases $2.4\%$. F2PS uniformly samples the point cloud and preserves the key information. Thus, the performance of semantic segmentation has been improved.

## 4.6 Comparision with Restoring-Based Methods

Based on the same baseline network in Sec. 4.5, we compare our SFCNet with the methods that compensate for the quantized information loss by restoring complete predictions from partial predictions, including the KNN-based post-processing [13] and KPConv refinement [17]. Tab. 4 shows that SFCNet has $3.2\%$ mIoU improvement to the KNN-based post-processing and $2.8\%$ mIoU improvement to KPConv refine-

Table 4: The performance comparison between the restoring-based methods and SFCNet on the SemanticKITTI validation set.

| Method | mIoU (%) |
|---|---|
| KNN-based Post-processing [13] | 59.7 |
| KPConv Refinement [17] | 60.1 |
| SFCNet (Ours) | **62.9** |

ment. Compared to the restoring-based methods, SFCNet preserves the complete geometric structure for the feature aggregation rather than compensating for the information loss by post-processing or refinement, which results in higher performance of semantic segmentation.

# 5 Conclusion

In this paper, we present the Spherical Frustum sparse Convolutional Network (SFCNet), a 2D convolution-based LiDAR point cloud segmentation method without quantized information loss. The quantized information loss is eliminated through the novel spherical frustum structure, which preserves all the points projected onto the same 2D position. Moreover, the novel spherical frustum sparse convolution and frustum farthest point sampling are proposed for effective convolution and sampling of the points stored in the spherical frustums. Experiment results on SemanticKITTI and nuScenes datasets show the better semantic segmentation performance of SFCNet compared to the previous 2D projection-based semantic segmentation methods, especially on small objects. The results show the great potential of SFCNet for safe autonomous driving perception due to the accurate segmentation of small targets.

**Limitations and future work.** To implement the 2D convolution on the spherical frustum, only the nearest points in the neighbor spherical frustums are adopted in the spherical frustum sparse convolution. This design may limit the receptive field of the network and thus result in a slightly weaker performance of the wide-range classes. To maintain the performance on both the wide-range classes and small classes, the improvement direction is to expand the receptive field based on our spherical frustum structure. To realize this, future work can lie in combining the vision network architecture with a larger receptive field, like the vision transformer [49] or vision mamba [50], with our spherical frustum structure. In addition, our work mainly focuses on the supervised and unimodal point cloud semantic segmentation. Future work can also lie in adopting the spherical frustum structure on the weakly-supervised [51] and multi-modal [52] point cloud semantic segmentation. Moreover, applying the spherical frustum structure to more tasks on the LiDAR point cloud, like point cloud registration [53, 54] and scene flow estimation [55], is also a direction for future work.

# 6 Acknowledgments and Disclosure of Funding

This work was supported in part by the Natural Science Foundation of China under Grant 62225309, 62073222, U21A20480 and 62361166632.

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

# Appendix

In the appendix, we first introduce the detailed architecture of the Spherical Frustum sparse Convolution Network (SFCNet) in Sec. A. Then, the additional implementation details of SFCNet are presented in Sec. B. Next, the additional experimental results are illustrated in Sec. C. Finally, more visualization of the semantic segmentation results on the SemanticKITTI [1] and nuScenes [27] datasets are presented in Sec. D.

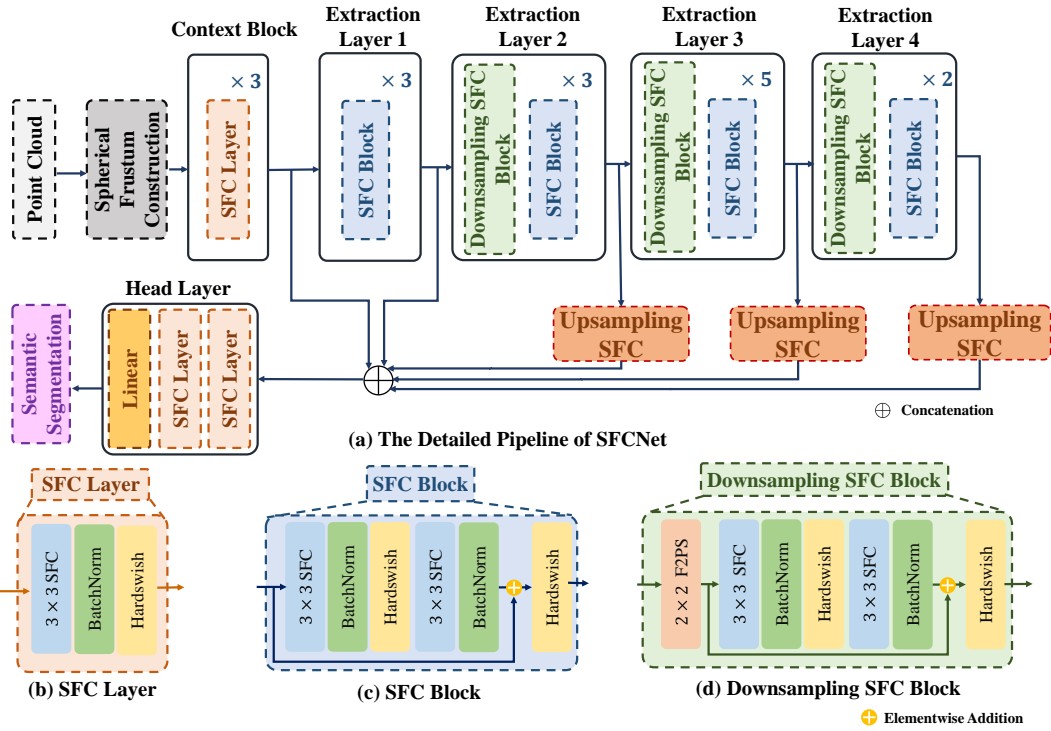

Figure 5: The Detailed Architecture of SFCNet. (a) presents the detailed pipeline of SFCNet. In addition, (b), (c), and (d) show the detailed module structures of the SFC layer, SFC block, and downsampling SFC block respectively, where SFC means spherical frustum sparse convolution, and F2PS means the frustum farthest point sampling.

## A    Detailed Architecture

Fig. 5 shows the detailed architecture of SFCNet. In SFCNet, the spherical frustum structure of the input point cloud is first constructed. Then, the encoder, which consists of the context block and extraction layers 1 to 4, is adopted for the point feature extraction. Next, in the decoder, the point features extracted in extraction layers 2 to 4 are upsampled by the upsampling Spherical Frustum sparse Convolution (SFC). The upsampled features are concatenated with the features extracted in the context block and the extraction layer 1. The concatenated features are fed into the head layer to decode the point features into the semantic predictions.

In addition, Fig. 5 also shows the three basic modules in SFCNet, including the SFC layer, SFC block, and downsampling SFC block. Moreover, we present the detailed hyperparameters of SFCNet in Tab. 5.

**Basic Modules of SFCNet.**    Specifically, the SFC layer is composed of the SFC, batch normalization, and the activation function. Inspired by [20], we use Hardswish [56] as the activation function. The formula of Hardswish is:

$$Hardswish(x) = \begin{cases} 0 & if\ x \leq -3 \\ x & if\ x \geq 3 \\ x \cdot (x+3)/6 & otherwise \end{cases} . \tag{3}$$

Table 5: The Detailed Hyperparameters of the Components and Modules in SFCNet. Component shows the names of components in SFCNet. Module Type shows the types of basic modules used in the components. Kernel Size shows the convolution kernel size of Spherical Frustum sparse Convolutions (SFCs) used in the modules. In the column of Stride, $[1,1]$ strides mean the SFC treats all the points as the center points, while $[2,2]$ strides show the strides used in Frustum Farthest Point Sampling (F2PS). The column of Upsampling Rate shows the upsampling rate used in the upsampling SFCs. Number of Modules shows the number of composed modules used in corresponding components. Width shows the output channel dimensions of the modules. In addition, in the column of Width, $C$ means the channel dimensions of the extracted point features, which is 128 for the SemanticKITTI dataset and 256 for the nuScenes dataset. $n$ means the number of semantic classes, which is 19 for the SemanticKITTI dataset and 16 for the nuScenes dataset.

| Component | Module Type | Kernel Size | Stride | Upsampling Rate | Number of Modules | Width |
|---|---|---|---|---|---|---|
| Context Block | SFC Layer | [3,3] | [1,1] | — | 3 | [C/2,C,C] |
| Extraction Layer 1 | SFC Block | [3,3] | [1,1] | — | 3 | [C,C,C] |
| Extraction Layer 2 | Downsampling SFC Block | [3,3] | [2,2] | — | 1 | [C] |
| | SFC Block | [3,3] | [1,1] | — | 3 | [C,C,C] |
| Extraction Layer 3 | Downsampling SFC Block | [3,3] | [2,2] | — | 1 | [C] |
| | SFC Block | [3,3] | [1,1] | — | 5 | [C,C,C,C,C] |
| Extraction Layer 4 | Downsampling SFC Block | [3,3] | [2,2] | — | 1 | [C] |
| | SFC Block | [3,3] | [1,1] | — | 2 | [C,C] |
| Upsampling | Upsampling SFC for Extraction Layer 2 | [3,3] | [1,1] | [2,2] | 1 | [C] |
| | Upsampling SFC for Extraction Layer 3 | [7,7] | [1,1] | [4,4] | 1 | [C] |
| | Upsampling SFC for Extraction Layer 4 | [15,15] | [1,1] | [8,8] | 1 | [C] |
| Head Layer | SFC Layer | [3,3] | [1,1] | — | 2 | [2·C,C] |
| | Linear | — | — | — | 1 | [n] |

The SFC block consists of two SFC layers. In addition, the residual connection [44] is adopted in the SFC block to overcome network degradation.

The downsampling SFC block combines the downsampling of Frustum Farthest Point Sampling (F2PS) and the feature aggregation of the SFC block. Notably, in the downsampling SFC block, the first SFC treats the sampled points as the center points and the features of the point cloud before sampling as the aggregated features.

Moreover, after the downsampling, the 2D coordinates of each spherical frustum are divided by the stride to gain the 2D coordinates on the downsampled 2D spherical plane. Meanwhile, each point is assigned a new point index in the downsampled spherical frustum point set according to the sampled order in F2PS.

**Components in the Encoder of SFCNet.** In the encoder, the context block consists of three SFC layers to extract the initial point features from the original point cloud. The subsequent four extraction layers are composed of $3, 3, 5,$ and $2$ SFC blocks respectively. In addition, a downsampling SFC block with $(2, 2)$ strides is adopted in the last three layers to downsample the point cloud into different scales. Thus, the multi-scale point features are extracted.

**Components in the Decoder of SFCNet.** We implement the upsampling SFC in the decoder of SFCNet according to the deconvolution [57] used in the 2D convolutional neural networks. In the upsampling SFC, we first multiply the 2D coordinates of the spherical frustums in the corresponding layer by the upsampling rate to obtain the 2D coordinates on the original spherical plane. Then, each point in the raw point cloud is treated as the center point in SFC. The spherical frustums fall in the convolution kernel are convolved. As shown in Tab. 5, we set the appropriate kernel size according to the upsampling rate for each upsampling SFC.

After the upsampling, the point features from different extraction layers are of the same size. Thus, the point features can be concatenated. In the head layer, two SFC layers and a linear layer are adopted for the decoding of the concatenated features.

# B    Additional Implementation Details

**Data Normalization.**    For the $k$-th point in the LiDAR point cloud $\mathcal{P}$, the combination of the 3D coordinates $\boldsymbol{x}_k = [x_k, y_k, z_k]^T$, the range $r_k = \sqrt{x_k^2 + y_k^2 + z_k^2}$, and the intensity is treated as the input point feature $\boldsymbol{f}_k$. Because of the different units of the different data categories, the input features should be normalized.

Specifically, for the SemanticKITTI [1] dataset, like RangeNet++ [13], we minus the features by the mean and divide the features by the standard deviation to obtain the normalized features. The mean and standard deviation are obtained from the statistics of each input data category on the SemanticKITTI dataset, which are presented in Tab. 6.

Table 6: The statistics of each input data category on SemanticKITTI dataset.

| Statistics | $x$ | $y$ | $z$ | $range$ | $intensity$ |
|---|---|---|---|---|---|
| Mean | 10.88 | 0.23 | -1.04 | 12.12 | 0.21 |
| Standard Deviation | 11.47 | 6.91 | 0.86 | 12.32 | 0.16 |

For the nuScenes [27] dataset, like Cylinder3D [23], a batch normalization layer is applied on the input point features to record the mean and standard deviation of the nuScenes dataset during training. During inferencing, the recorded mean and standard deviation are used to normalize the input point features.

**Spherical Frustum Construction.**    We construct the spherical frustum structure by assigning each point with the 2D spherical coordinates $(u_k, v_k)$ and the point index $m_k$ in the spherical frustum point set, where $k$ is the index of the point in the original point cloud. The 2D spherical coordinates can be calculated through Eq. 1. Thus, the key process is to assign the point index $m_k$ for each point based on the 2D spherical coordinates.

We implement this by sorting the 2D coordinates $(u_k, v_k)$ of the points. The points with smaller $u_k$ and $v_k$ are ranked ahead of the points with larger $u_k$ and $v_k$. Thus, the points with the same 2D coordinates are neighbors in the sorted point cloud. For each point, we count the number of points that have the same 2D coordinates and appear ahead or behind the point in the sorted point cloud separately. The number of the points appearing ahead is treated as the point index $m_k$ of each point.

In addition, we assign each point an indicator $\xi_k \in \{0, 1\}$ according to the number of the points appearing behind. The point with zero point appearing behind is assigned a zero indicator. Otherwise, the point is assigned with an indicator equal to one. The indicator indicates the end of the frustum point set and is used for the subsequent spherical frustum point set visiting.

The sorting and the point number counting are implemented through the Graphics Processing Unit (GPU)-based parallel computing using Compute Unified Device Architecture (CUDA). Thus, the construction is efficient in practice.

**Hash-Based Spherical Frustum Representation.**    After the construction of the spherical frustum structure, we build the hash-based spherical frustum representation. Specifically, we construct the key-value pairs between the key $(u_k, v_k, m_k)$ and the value $k$. The key-value pairs are inserted into a hash table, which represents the neighbor relationship of spherical frustums and points.

In practice, we adopt an efficient GPU-based hash table [58]. The GPU-based hash table requires both key and value to be an integer. The value $k$ satisfies the integer requirement. However, the key $(u_k, v_k, m_k)$ in the hash-based spherical frustum representation is not an integer.

To adopt the GPU-based hash table for efficient processing, $(u_k, v_k, m_k)$ is transferred to an integer as $v_k \cdot (W \cdot M) + u_k \cdot M + m_k$, where $W$ is the width of the spherical projection, $M$ is the maximal point number of the spherical frustum point sets. Through this process, any point represented by the coordinates $(u_k, v_k, m_k)$ can be efficiently queried through the GPU-based hash table.

**Spherical Frustum Point Set Visiting.**    Both the SFC and F2PS require visiting all the points in any spherical frustums. Thus, we propose the spherical frustum point set visiting algorithm. The visiting obtains all the points in the given spherical frustum, whose 2D coordinates are $(u, v)$, by sequentially querying the points in the hash table.

Specifically, we first query the first point in the spherical frustum using the key $(u, v, 0)$. If the key $(u, v, 0)$ is not in the hash table, the spherical frustum on $(u, v)$ is invalid. Otherwise, the first point in the spherical frustum can be queried through the hash table.

Then, the points in the spherical frustum are sequentially visited. We first initialize the point index $m = 0$ in the spherical frustum. At each step, the point index $m$ increases by one. Through the hash table, the point with $m$-th index in the spherical frustum is queried using the key $(u, v, m)$. Meanwhile, the indicator $\xi$ of this point is obtained. $\xi$ indicates whether $(u, v, m + 1)$ refers to a valid point. Thus, the visiting ends when the indicator of the current point is zero.

**Detailed Implementation of Frustum Farthest Point Sampling.** In F2PS, we first sample the spherical frustums by stride. Then, we sample the points in each sampled spherical frustum by Farthest Point Sampling (FPS) [26]. As mentioned in Sec. 3.3, FPS is an iterative algorithm. The detailed process of the $j$-th iteration can be expressed by the following formula:

$$S_j = S_{j-1} \cup \{\arg \max_{p \in P_s \setminus S_{j-1}} \min_{s \in S_{j-1}} dist(p, s)\}, \tag{4}$$

where $P_s$ is the spherical frustum point set to be sampled, $S_j$ and $S_{j-1}$ are the sampled point sets in $j$-th and $(j-1)$-th iterations respectively. Notably, $S_0$ contains the point randomly sampled from $P_s$. In addition, $dist(p, s)$ is the distance between point $p$ and point $s$ in 3D space. The iteration starts at $j = 1$, and ends when the size of $S_k$ equals the number of sampling points.

Moreover, since the distances between the points in $S_{j-2}$ and the points in $P_s \setminus S_{j-1}$ have been calculated before the $j$-th iteration, we just need to calculate the distance between each $p$ in $P_s \setminus S_{j-1}$ and the point sampled in $(j-1)$-th iteration for the calculation of $\min_{s \in S_{j-1}} dist(p, s)$, which is the minimal distance from point $p$ to the point set $S_{j-1}$. Thus, the computing complexity of FPS for $P_s$ of size $n$ is $O(n^2)$. Since the point number of each spherical frustum is $O(1)$, the computing complexity of FPS for the spherical frustum is also $O(1)$, which ensures the efficiency of F2PS.

**Loss Function.** We use multi-layer weighted cross-entropy loss and Lovász-Softmax loss [47] to help the network learn the semantic information from different scales. To get the semantic predictions of extraction layers 1 to 4, we apply a linear layer to decode the extracted point features of each extraction layer into the semantic predictions.

Specifically, for extraction layer 1, the linear layer is applied on the extracted point features $\mathcal{F}_1$ to gain the prediction $\tilde{L}_1$. For the other extraction layers, the linear layer is applied on the upsampled point features $\mathcal{F}'_2$, $\mathcal{F}'_3$, and $\mathcal{F}'_4$ to obtain the predictions $\tilde{L}_2$, $\tilde{L}_3$, and $\tilde{L}_4$ respectively.

Based on the predictions of each layer and the final predictions of SFCNet $\tilde{L}_1$, the loss function is calculated as:

$$\mathcal{L} = \sum_{i=1}^{4} \mathcal{L}_{wce}(\tilde{L}_i, L) + \mathcal{L}_{Lov}(\tilde{L}_i, L), \tag{5}$$

where $\mathcal{L}_{wce}$ is the weighted cross-entropy loss, $\mathcal{L}_{Lov}$ is the Lovász-Softmax loss, and $L$ is the ground truth. In addition, the weights of weighted cross-entropy loss are calculated as $w_c = (f_c + \epsilon)^{-1}$, where $c$ is the semantic class, $f_c$ is the frequency of class $c$ in the dataset, and $\epsilon$ is a small positive value to avoid zero division.

## C   Additional Experiments

### C.1   Efficiency Comparison

We evaluate the efficiency of the proposed SFCNet with the previous works and our 2D projection-based baseline model on a single Geforce RTX 4090Ti GPU.

We adopt the same baseline model used in Sec. 4.5. For RangeViT, we adopt the official code for efficiency evaluation. Notably, in the inference, RangeViT splits the projected LiDAR image, inputs each image slice into the network to gain the predictions, and merges the predictions to gain the prediction of the entire projected LiDAR image. Thus, the inference time of RangeViT includes the time of all the processes. In addition, since RangeViT adopts the KPConv refinement [17], which restores the complete predictions from the partial predictions, we use the point number of the entire point cloud as the processed point number. For PointNet++ [26], we sample $45K$ points from the point cloud before inputting into the network as its original setting. For 3D voxel-based methods, Cylinder3D [23] and SphereFormer [25], only the points preserved after the voxelization are counted since these points are exactly processed in the 3D sparse convolution network.

Table 7: Efficiency comparison. The inference time of a single LiDAR scan, the processed point number, and the normalized time, the inference time per thousand points, are evaluated on the SemanticKITTI validation set with a single Geforce RTX 4090Ti GPU.

| Approach | Time (ms)/Points ↓ | Normalized Time (ms/$K$) ↓ |
|---|---|---|
| PointNet++ [26] | 131.0/$\sim 45K$ | 2.91 |
| RandLA [31] | 212.2/$\sim 120K$ | 1.77 |
| Cylinder3D [23] | 67.5/$\sim 40K$ | 1.69 |
| SphereFormer [25] | 108.2/$\sim 90K$ | 1.20 |
| RangeViT [21] | 104.8/$\sim 120K$ | 0.87 |
| Baseline | 46.4/$\sim 90K$ | 0.52 |
| SFCNet (Ours) | **59.7/$\sim$ 120K** | **0.49** |

Table 8: The quantitative results of different resolutions used in the baseline model with KNN-based post-processing [13] on the SemanticKITTI validation set.

| Resolution | Preserved Points/All Points | mIoU (%) ↑ |
|---|---|---|
| $64 \times 1800$ | $88K/120K$ | **59.7** |
| $64 \times 2048$ | $97K/120K$ | 58.9 |
| $64 \times 4096$ | $113K/120K$ | 57.0 |

The results are presented in Tab. 7. The results show that SFCNet costs 59.7 ms for a single scan inference, which reaches real-time LiDAR scan processing. In addition, our SFCNet also has the highest efficiency evaluated by normalized time (0.49 ms/$K$) compared to the previous 3D and 2D methods and the 2D baseline model, which indicates that SFCNet can adopt the 2D projection property to efficiently segment the large-scale point cloud.

## C.2 Analysis on Different Resolutions of the Baseline Model

Since the limited projection resolution is the reason for quantized information loss, expanding the resolution of the projected range image can preserve more points during the spherical projection and ease the quantized information loss. However, expanding the resolution increases the sparsity of the projected points and makes the convolution hard to aggregate the local features. Thus, resolution expansion is not a feasible solution for resolving quantized information loss. To validate this, we expand the image horizon resolution of the baseline model to 2048 and 4096 and conduct the ablation studies of different resolutions on the SemanticKITTI validation set to show the effect of a larger resolution. As shown in Tab. 8, the increment of resolution preserves more points but results in worse performances. In contrast, SFCNet not only overcomes quantized information loss but also effectively aggregates local features with a suitable resolution by preserving all points using spherical frustum.

## C.3 Additional Ablation Studies

In this subsection, we conduct additional ablation studies to evaluate the sensitivity of our SFCNet to the key parameters.

**Stride Sizes in Frustum Farthest Point Sampling (F2PS).** The ablation studies of four different settings of the stride sizes in the F2PS on the three downsampling layers are conducted, including $(1, 2), (2, 1), (2, 4)$, and $(4, 2)$. The results are shown in Tab. 9. The results show on all the downsampling layers, the $(2, 2)$ stride sizes show a better segmentation performance than the other stride size settings. $(2, 2)$ stride sizes suitably downsample the point cloud in the vertical and horizon dimensions. Higher or lower downsampling rates result in the oversampling or undersampling of the point cloud respectively.

**Number of Points in the Spherical Frustums.** In the spherical frustum structure, the number of points in the frustum is unlimited and only depends on how many points are projected onto the corresponding 2D location. To analyze the effect of the number of points in the frustum, we set the maximal number of points in each spherical frustum and the points exceeding the maximal point number are dropped. As shown in Tab. 10, preserving more points in the spherical frustum results in better segmentation performance, since more complete geometry information is preserved. These

Table 9: Ablation study on the stride sizes of the Frustum Farthest Point Sampling in the downsampling layers on the SemanticKITTI validation set.

| Stride Sizes $(S_h, S_w)$ | mIoU (%) ↑ | | |
| --- | --- | --- | --- |
| | Layer 1 | Layer 2 | Layer 3 |
| (2,1) | 60.7 | 61.3 | 61.1 |
| (1,2) | 62.4 | 62.2 | 62.3 |
| (2,4) | 62.3 | 62.6 | 61.9 |
| (4,2) | 60.5 | 61.6 | 61.9 |
| (2,2) (Ours SFCNet) | **62.9** | | |

Table 10: Ablation study on the maximal number of points in spherical frustums on the SemanticKITTI validation set.

| Maximal Number of Points in Spherical Frustum | mIoU (%) ↑ |
| --- | --- |
| 2 | 61.0 |
| 4 | 61.9 |
| Unlimited (Ours SFCNet) | **62.9** |

results further indicate the significance of overcoming quantized information loss in the field of LiDAR point cloud semantic segmentation.

**Configuration of the Hash Table.** The number of hash functions is the main parameter of the hash table, which means the number of functions used for the hash table retrieval. In the implementation, if the first hash function can successfully retrieve the location of the target point, the other functions will not be used. We change the number of hash functions to show the model sensitivity of hash table configurations. As shown in Tab. 11, the performance and inference time of SFCNet have little difference under different numbers of hash functions. The results show that in most cases, the first function can successfully retrieve the location, and thus the inference times change slightly in different function numbers. These results indicate that SFCNet is robust to different hash table configurations.

## C.4 Comparison of Sampling Methods

We further validate the effectiveness and efficiency of the proposed Frustum Farthest Point Sampling (F2PS) by the qualitative comparison with stride-based 2D sampling and the comparison of time consumption with Farthest Point Sampling (FPS).

**Qualitive Comparison.** As shown in Fig. 6(a), Stride-Based 2D Sampling (SBS) only samples the point cloud based on 2D stride. The visualization shows that the stride-based sampled point cloud is relatively rough. Due to the lack of 3D geometric information, SBS fails to sample the 3D point cloud uniformly. Thus, the loss of geometric structure in the sampled point cloud is obvious, such as many broken lines on the ground. Our F2PS takes into account the 3D geometric information based on the FPS in the spherical frustum, which enables F2PS to sample the 3D point cloud uniformly and preserve the significant 3D geometric structure during the sampling.

**Time Consumption Comparison.** As shown in Fig. 6(b), with the increment of sampled point number, the cost time of our F2PS increases slowly, while the cost time of FPS increases dramatically. This result shows performing FPS on the frustum point sets is efficient and does not increase the computing burden.

## C.5 Comparison between SFCNet and Baseline Model on Small Object Categories

To further show the improvement of small object segmentation, we compare the quantitative results on the small object categories between SFCNet and the baseline model with the KNN-based post-processing [13] on SemanticKITTI and nuScenes validation sets. As shown in Tabs. 12 and 13, our SFCNet has higher performances on all small object categories compared to the baseline model. The results show overcoming quantized information loss preserves complete geometric information of the small objects and thus makes them better recognized and segmented by our SFCNet.

Table 11: Ablation study on the configuration of the hash table for the spherical frustum structure on the SemanticKITTI validation set.

| Number of Hash Functions | Inference Time (ms) ↓ | mIoU (%) ↑ |
|---|---|---|
| 2 | 59.5 | 62.9 |
| 3 | 60.1 | 62.9 |
| 5 | 59.5 | 62.9 |
| 4 (Ours SFCNet) | 59.7 | 62.9 |

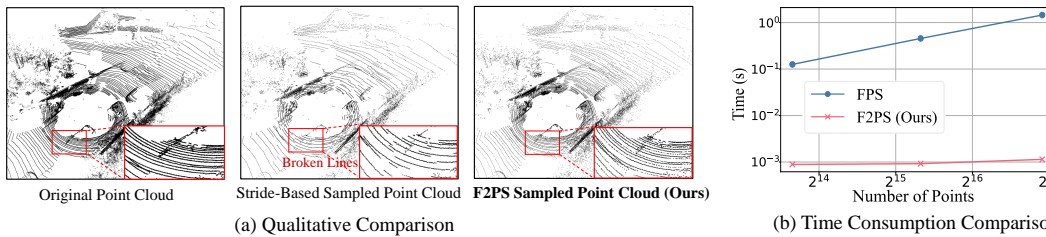

(a) Qualitative Comparison          (b) Time Consumption Comparison

Figure 6: In this figure, (a) presents the qualitative comparison between stride-based 2D sampling and our Frustum Farthest Point Sampling (F2PS). The red boxes show the zoomed view of the point clouds in the close areas. (b) illustrates the time consumption comparison between Farthest Point Sampling (FPS) and our F2PS.

## D   More Visualization

To better demonstrate the effectiveness of SFCNet for LiDAR point cloud semantic segmentation, we conduct more visualization on the SemanticKITTI and nuScenes datasets. The results are shown in Figs. 7, 8, 9, 10 and the supplementary video SupplementaryVideo.mp4.

**Qualitative Comparison on NuScenes Validation Set.**   The results of qualitative comparison between our SFCNet and RangeViT [21] are shown in Fig. 7. On the nuScenes validation set, SFCNet can also have fewer segmentation errors than RangeViT as the results in the SemanticKITTI dataset. Moreover, the better segmentation accuracy of the 3D small objects, like pedestrians and motorcycles, can also be observed on the nuScenes validation set. The results once more demonstrate semantic segmentation improvement of SFCNet due to the overcoming of quantized information loss.

**More Qualitative Comparison on SemanticKITTI Test Set.**   The ground truths on the SemanticKITTI test set are not available. Thus, we search for the corresponding RGB image and project the semantic predictions on the image to compare the semantic segmentation accuracy between the state-of-the-art 2D image-based method CENet [20] and our SFCNet on the SemanticKITTI test set. As shown in Figs. 8 and 9, compared to CENet, SFCNet can more accurately segment the LiDAR point cloud in various challenging scenes on the SemanticKITTI test set.

Specifically, SFCNet recognizes the thin poles in distance on the rural road of Fig. 8(a) and in the complex intersections of Fig. 9(c), while CENet predicts the poles as wrong classes. In addition, SFCNet recognizes the thin trunks inside the vegetation on the rural scenes of Fig. 8(b) and Fig. 9(b) while CENet wrongly predicts the trunk as the fetch and vegetation respectively. Moreover, SFCNet successfully segments the boxed persons in the complex intersection of Fig. 8(c) and in the urban scene of Fig. 9(a) while CENet gives wrong predictions due to the information loss of the distant persons during 2D projection. These results further validate the better segmentation performance of SFCNet to 3D small objects.

**More Qualitative Comparison on NuScenes Validation Set.**   As the visualization on the SemanticKITTI test set, we provide the additional qualitative comparison between our SFCNet and RangeViT on the nuScene validation set with the projected predictions illustrated in Fig. 10. The results further demonstrate the better semantic segmentation of SFCNet for the challenging street scenes on the nuScenes validation set compared to RangeViT.

Specifically, in the first scene, the close motorcycle can be correctly segmented by SFCNet, while RangeViT recognizes the motorcycle as a car, which shows that SFCNet can help the autonomous car correctly recognize the type of close obstacles, and enable the car to make appropriate decisions.

Table 12: Quantative comparison of semantic segmentation between baseline model and SFCNet for the small object categories on SemanticKITTI validation sets. **Bold** results are the best in each column. The performance improvement of each category is highlighted in green.

| Approach | SemanticKITTI | | | | | | | |
| | bicycle | motorcycle | person | bicyclist | motorcyclist | trunk | pole | traffic-sign |
| --- | --- | --- | --- | --- | --- | --- | --- | --- |
| Baseline w/ KNN-Based Post-processing | 44.2 | 46.0 | 48.8 | 71.6 | 73.6 | 67.4 | 63.1 | 45.7 |
| SFCNet (Ours) | **44.9**(+0.7) | **60.6**(+14.6) | **50.5**(+1.7) | **73.1**(+1.5) | **83.1**(+9.5) | **68.5**(+1.1) | **64.6**(+1.5) | **47.8**(+2.1) |

Table 13: Quantative comparison of semantic segmentation between baseline model and SFCNet for the small object categories on the nuScenes validation sets. **Bold** results are the best in each column. The performance improvement of each category is highlighted in green.

| Approach | nuScenes | | | |
| | bicycle | motorcycle | pedestrian | traffic-cone |
| --- | --- | --- | --- | --- |
| Baseline w/ KNN-Based Post-processing | 30.6 | 77.0 | 73.9 | 62.8 |
| SFCNet (Ours) | **40.4**(+9.8) | **82.0**(+5.0) | **78.0**(+4.1) | **65.8**(+3.0) |

In the second scene, the distant pedestrians on the other side of the crossing can also be correctly segmented by SFCNet due to the elimination of quantized information loss. In contrast, RangeViT wrongly predicts the pedestrians as traffic cones.

In the third scene, since the boxed pedestrian is close to the manmade, RangeViT confuses it with the manmade and does not segment the pedestrian, while our SFCNet can clearly recognize the boundary and successfully segments the pedestrian.

**Sequential Qualitative Comparison on SemanticKITTI Validation Set.** We demonstrate the qualitative comparison between our SFCNet and the SoTA 2D projection-based segmentation methods, CENet and RangeViT, on a continuous sequence on the SemanticKITTI validation set in the supplementary video `SupplementaryVideo.mp4`. In this video, the semantic predictions in both the 3D point cloud view and the RGB image view (where the colored point cloud is projected onto the RGB images) are presented. The results show that our SFCNet can consistently show higher segmentation accuracy on the point cloud of each frame in the sequence than 2D projection-based methods, which further indicates the stronger semantic segmentation capability of our SFCNet.

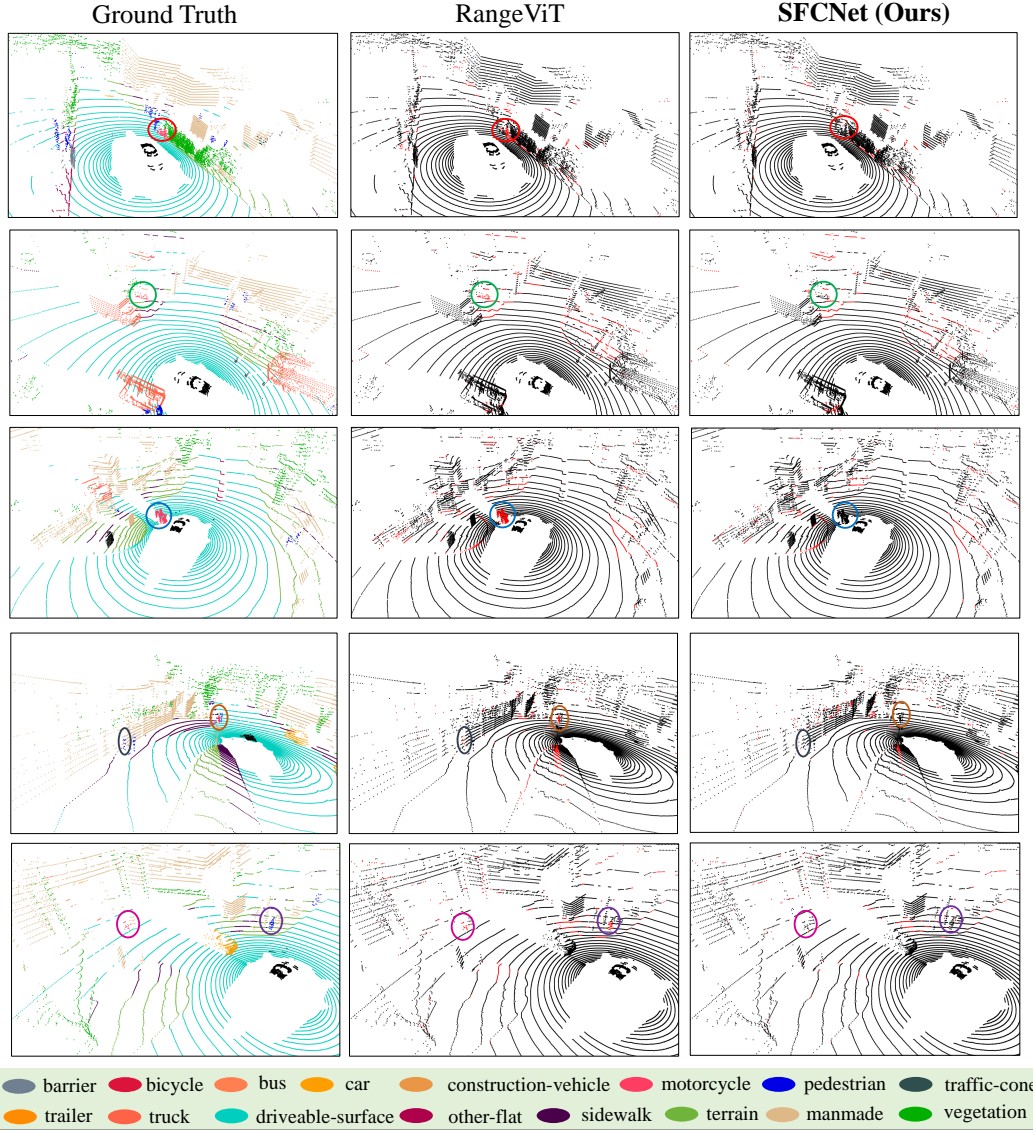

Figure 7: Qualitative Comparison on NuScene Validation Set. In this figure, we conducted the qualitative comparison between RangeViT [21] and our SFCNet of semantic segmentation on the nuScenes validation set. The first column presents the ground truths, while the following two columns show the error maps of the predictions of RangeViT and our SFCNet respectively. In addition, the reference from point color to the semantic class in the ground truths is shown at the bottom. Moreover, the false-segmented points are marked as red in the error maps. Furthermore, we use circles with the same color to point out the same objects in the ground truth and the two error maps.

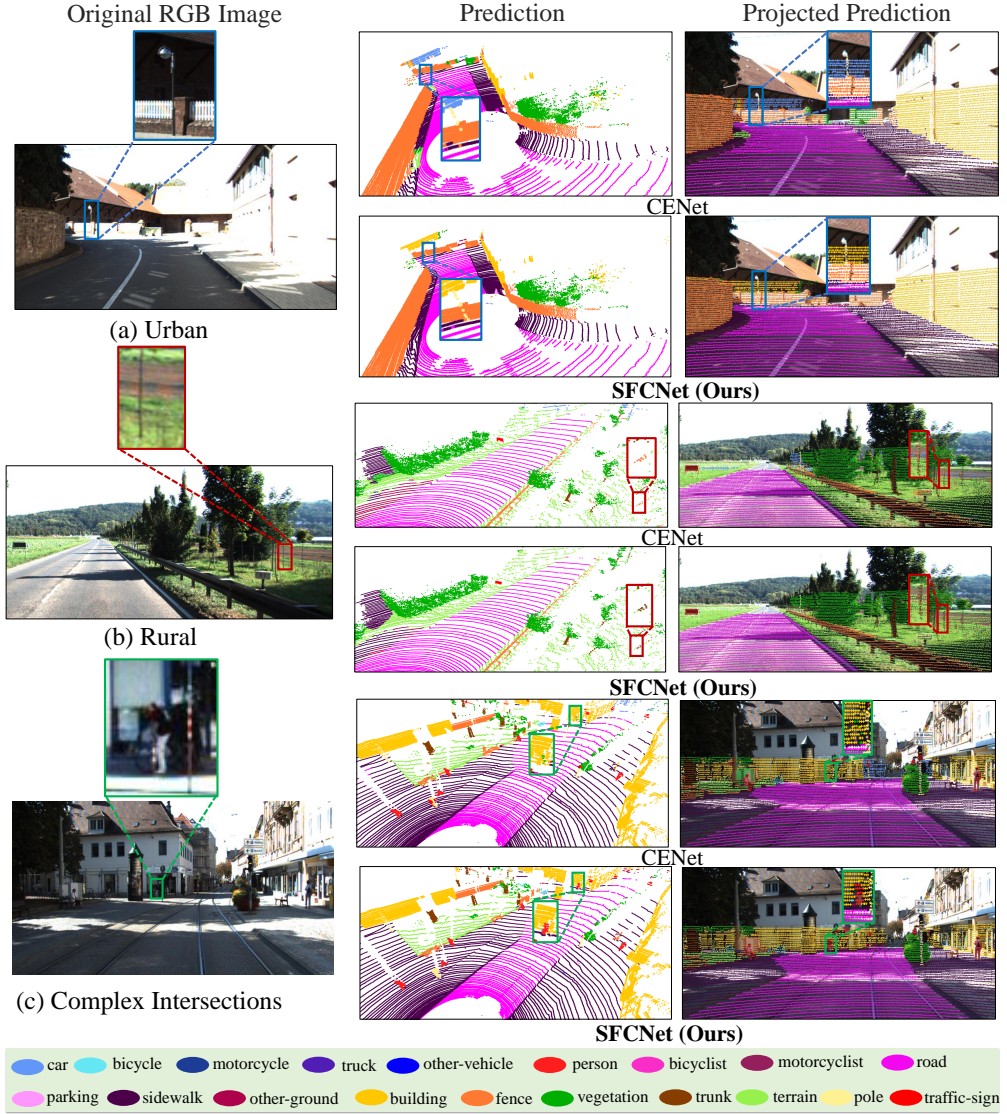

Figure 8: More Qualitative Comparison on Semantic Segmentation on SemanticKITTI Test Set. We show the qualitative comparison between our SFCNet and the state-of-the-art 2D image-based method CENet [20] on the SemanticKITTI test set. The visualized challenging autonomous driving scenes include urban, rural, and complex intersection scenes. The predictions projected on the corresponding RGB images are also illustrated. In addition, we use the same color boxes to point out the same objects in the point clouds and images for each scene. Meanwhile, we provide the zoomed-in view of some boxed objects for clear visualization. Moreover, the reference from point color to the semantic class in the predictions is shown at the bottom of the figure.

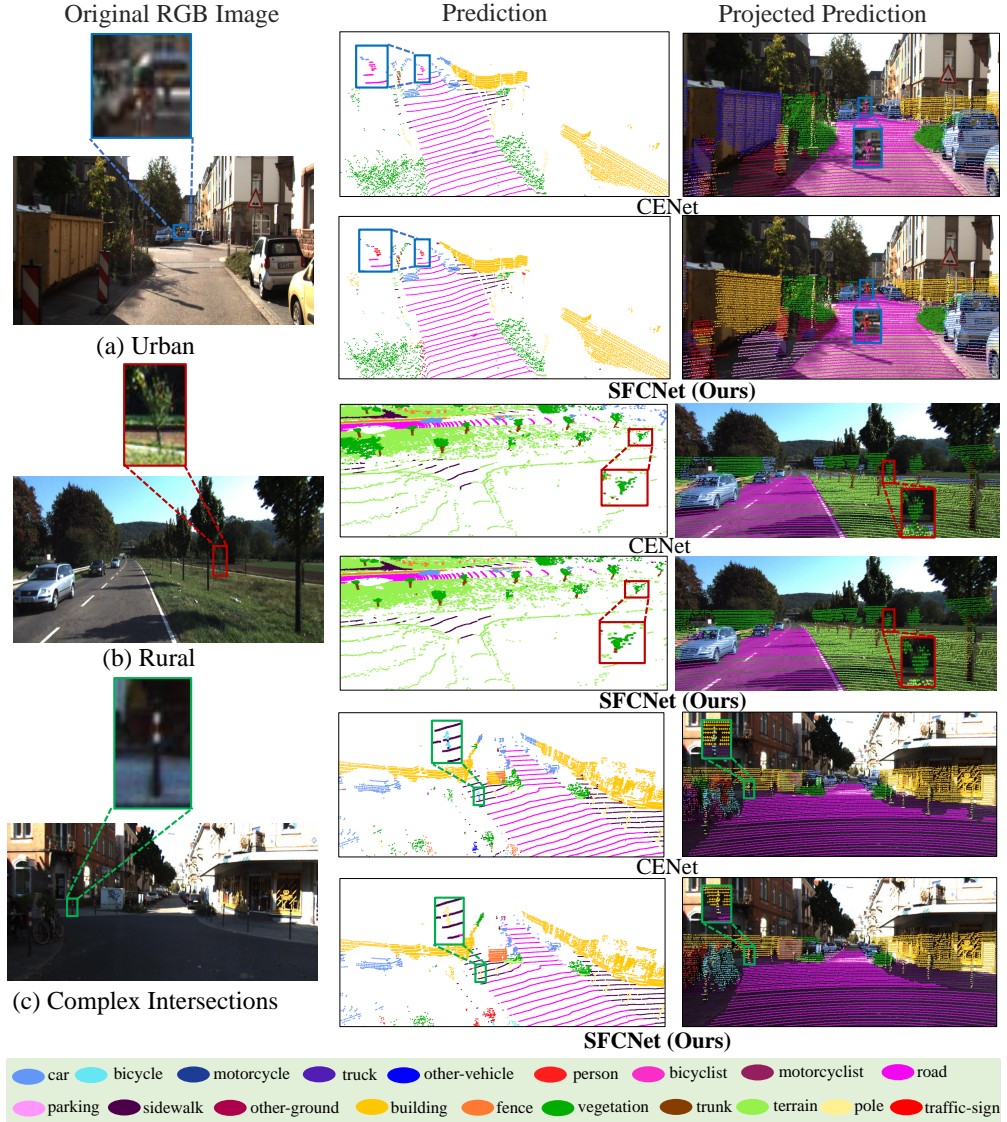

Figure 9: More Qualitative Results on Semantic Segmentation on SemanticKITTI Test Set. This figure shows more qualitative results of our SFCNet and the state-of-the-art 2D image-based method CENet [20] on the urban, rural, and complex intersection scenes of the SemanticKITTI test set. As in Fig. 8, the predictions projected on the corresponding RGB images are also illustrated. In addition, the same color boxes are adopted to point out the same objects in the point clouds and images for each scene. Meanwhile, the zoomed-in view of some boxed objects is illustrated for clear visualization. Moreover, the reference from point color to the semantic class in the predictions is shown at the bottom of the figure.

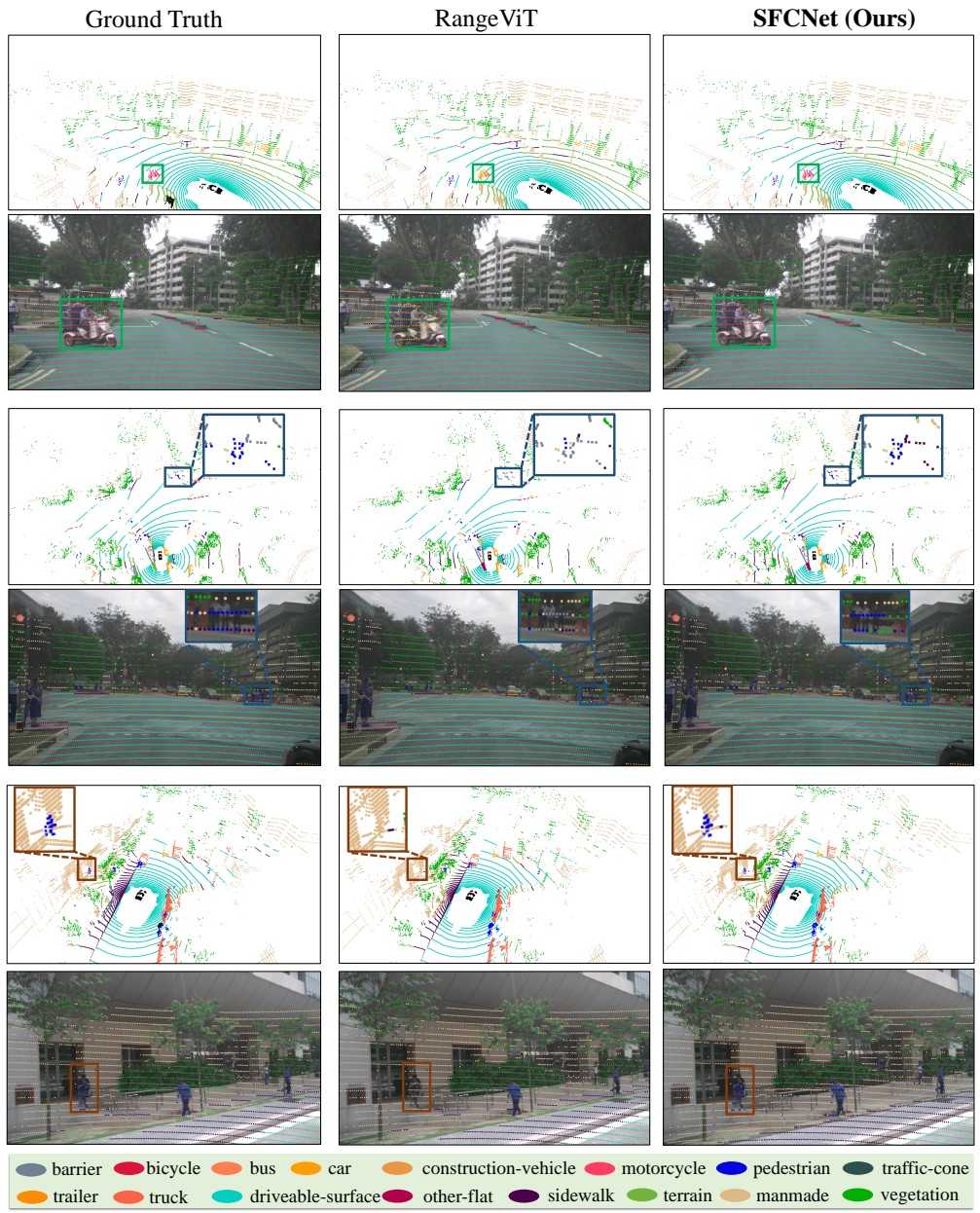

Figure 10: More Qualitative Comparison of Semantic Segmentation on NuScenes Validation Set. We show more comparisons between our SFCNet and the state-of-the-art 2D image-based method RangeViT [21] on the nuScenes dataset. The predictions projected on the corresponding RGB images are also illustrated. In addition, we use the same color boxes to point out the same objects in the point clouds and images for each scene. Meanwhile, we provide the zoomed-in view of some boxed objects for clear visualization. Moreover, the reference from point color to the semantic class in the predictions and ground truths is shown at the bottom of the figure.

