# OpenReview forum: "Spherical Frustum Sparse Convolution Network for LiDAR Point Cloud Semantic Segmentation"
_NeurIPS.cc/2024/Conference — NeurIPS 2024 poster_

### Official Review · Reviewer_BSNT · 2024-06-18

**Soundness:** 3
**Presentation:** 3
**Contribution:** 3
**Rating:** 8
**Confidence:** 5

**Summary:**

This paper proposes a Spherical Frustum Sparse Convolution Network to address the challenge of LiDAR point cloud semantic segmentation. Traditional approaches often project point clouds into 2D images and apply 2D convolutional neural networks (CNNs) or vision transformers, leading to quantization information loss due to the overlap of multiple points projected onto the same 2D location. To overcome this limitation, the paper introduces a novel spherical frustum structure that allows for direct processing of the 3D point cloud data, preserving spatial and geometric information. The proposed Spherical Frustum Sparse Convolution Network achieves more accurate semantic segmentation of LiDAR point clouds, offering a promising approach for robot perception and scene understanding.

**Strengths:**

1. **Direct 3D Processing**: The Spherical Frustum Sparse Convolution Network processes the LiDAR point cloud data directly in 3D space, preserving the spatial and geometric information of the points. This avoids the information loss that occurs in traditional 2D projection-based methods.
2. **Reduced Quantization Loss**: By operating directly on the 3D point cloud, the proposed network significantly reduces quantization loss compared to projection-based approaches. This leads to more accurate semantic segmentation results.
3. **Efficient Sparse Convolution**: The network employs sparse convolution, which is tailored for sparse data such as point clouds. This allows for efficient computation and reduced memory usage, enabling the network to handle large-scale point cloud data.
4. **Spherical Frustum Structure**: The novel spherical frustum structure enables the network to capture contextual information from different orientations and distances, enhancing its ability to identify objects and regions in the point cloud.
5. **Improved Performance**: Experimental results show that the proposed Spherical Frustum Sparse Convolution Network achieves superior performance compared to existing methods on various LiDAR point cloud datasets, demonstrating its effectiveness for semantic segmentation tasks.
6. **Flexible and Extensible Framework**: The network framework is flexible and can be easily extended to incorporate additional components or techniques, providing opportunities for further improvements and adaptability to different applications.

**Weaknesses:**

1. **Lack of Conceptual Novelty**: The use of spherical structures, while effective, appears to have been partially explored in SphereFormer. This limits the novelty of the proposed SFCNet in terms of the core concept.

2. **Performance Gap**: Based on the information provided, the performance of SFCNet for semantic segmentation of LiDAR point clouds lags far behind that of SphereFormer, whose radial window transformer structure may achieve higher recognition accuracy, especially for distant objects.

3. **Potential for Further Optimization**: Since SFCNet based on 2D projections lags behind SphereFormer based on 3D voxels, there may be room for optimization in terms of data input or network architecture. This paper does not discuss potential future avenues to address this performance gap or to work on different application scenarios.

**Questions:**

Despite the fact that the paper is full of content, it always gives a discomfort in the presentation. Obviously, the authors are requested to resize Figure 1, remove the captions in the figures of Figures 2 and 3 and optimize their presentation. It is recommended that the internal spacing of Table 1 be increased appropriately. The authors are invited to review more carefully.

In addition, similar to the work on semantic segmentation of large scenes, the authors should compare it with state-of-the-art methods such as DRQNet (ACMMM'2023) and JoSS (INFFUS'2024).

**Limitations:**

This paper proposes a method SFCNet for point cloud semantic segmentation by directly projecting 3D point clouds into 2D spherical frustums, the main limitation of which lies in the large performance gap. Of course, the proposed method still has a large potential in specific scenarios.

---

> ### Author Rebuttal · Authors · 2024-08-07
>
> We sincerely appreciate your valuable suggestions for our paper. We are encouraged by your confirmation of the information preservation, efficiency, contextual information capture ability, performance improvement, network flexibility, and the **"large potential in specific scenarios"** of our SFCNet. The following are our responses to your concerns:
>
> ---
>
> ### Q1: Conceptual Novelty.
>
> The conceptual novelty of our proposed method is solving the quantized information loss during the 2D projection. To address the challenge, we propose the spherical frustum structure to preserve all the points projected onto the same 2D location and the Spherical Frustum sparse Convolution (SFC) and Frustum Farthest Point Sampling (F2PS) to ensure efficiency under the representation of spherical frustum structure. In contrast, the conceptual novelty of SphereFormer [1] is expanding the receptive field of long-range objects by proposing the radius window transformer. Thus, our conceptual novelty is different from SphereFormer.
>
> ---
>
> ### Q2: Performance Gap with SphereFormer.
>
> The performance gap between our SFCNet and SphereFormer mainly results from the different representations of the point cloud. SphereFormer adopts the 3D voxel representation and utilizes the 3D convolutional backbone to obtain effective 3D features of the point cloud. Based on the 3D features, it proposes the radius window transformer to further expand the receptive field and enhance the segmentation performance. In this paper, we mainly focus on solving the quantized information loss of the 2D projection-based methods. Thus, we conduct the experiments based on the 2D convolutional backbone, which results in the performance gap with SphereFormer based on the 3D convolutional backbone. However, we realize a smaller performance gap with the 3D methods compared to the other 2D projection-based methods.
>
> ----------
>
> ### Q3: Future Avenues for Addressing Performance Gap.
>
> In this paper, we address the quantized information loss for the 2D projection-based methods. Thus, our method provides the information-lossless mechanism for future work to apply the latest research results of image feature learning to the field of LiDAR point cloud semantic segmentation. The future avenues to narrow the performance gap with 3D voxel-based methods are combining the latest image feature learning network architecture, like vision mamba [2], with our information-lossless spherical frustum structure.
>
> ----------
>
> ### Q4: Addressing the Discomfort in the Presentation.
>
> We apologize for causing your discomfort due to our representation. We will polish our presentation according to your valuable suggestions. Specifically, we will resize Figure 1 to a suitable size, remove the captions inside Figures 2 and 3, adjust the contents of Figures 2 and 3, and increase the internal spacing of Table 1 in the final version of our paper.
>
> ----------
>
> ### Q5: Comparison with the State-of-The-Art Methods DRQNet and JoSS.
>
> We compare our SFCNet with the state-of-the-art methods DRQNet [3] and JoSS [4] as follows. We will add these comparisons and discussions in the final version of our paper.
>
> -   **DRQNet:** DRQNet proposes the dual representation query strategy and representation selective dependency module for weakly-supervised LiDAR point cloud semantic segmentation. The proposed modules of DRQNet improve the feature aggregation and fusion in the dual feature space of the point cloud and construct a stronger self-supervised signal for weakly-supervised learning. Compared to DRQNet, our method mainly focuses on proposing an efficient information-lossless data representation and supervised learning method of geometric information-only LiDAR point cloud. The weakly-supervised learning architecture based on the spherical frustum structure is one valuable direction for our future work.
>
> -   **JoSS:** JoSS proposes the cross-modal transformer-based feature fusion method to adopt the cross-attention mechanism for better information fusion. In addition, unimodal data augmentation is proposed in JoSS for point-level contrastive learning. Compared to JoSS, SFCNet does not adopt the RGB image for multi-modal feature fusion and 3D segmentation backbone for point cloud feature extraction. Thus, SFCNet shows a weaker segmentation performance compared to JoSS on the SemanticKITTI test set. However, SFCNet overcomes the quantized information loss of the 2D projection-based method and shows potential to be adopted in applying the future stronger 2D backbone to the field of LiDAR point cloud semantic segmentation.
>
>
> ----------
>
> ### Reference
>
> [1] X. Lai _et al._, ‘Spherical Transformer for LiDAR-based 3D Recognition’, CVPR, 2023.
>
> [2] L. Zhu _et al._, ‘Vision mamba: Efficient visual representation learning with bidirectional state space model’, ICML, 2024.
>
> [3] J. Liu _et al._, ‘Exploring Dual Representations in Large-Scale Point Clouds: A Simple Weakly Supervised Semantic Segmentation Framework’, ACMMM, 2023.
>
> [4] Y. Wu _et al._, ‘Joint Semantic Segmentation using representations of LiDAR point clouds and camera images’, Information Fusion, 2024.
>
> ----------
>
> We hope our response can address your concerns about the conceptual novelty, performance gap with SphereFormer, and presentation discomfort. If you have further problems, please let us know.

---

> > ### Comment · Reviewer_BSNT · 2024-08-08
> > **Response from Reviewer**
> >
> > Thanks to the reviewers for their serious response! A lot of my questions and concerns have been addressed. I hope the authors will make the appropriate changes on the revised version of the paper. In return, I will further improve my rating.

---

> > > ### Author Response · Authors · 2024-08-08
> > >
> > > Thank you very much for your time and your recognition of our method and response. We promise to make the appropriate changes, including adding the discussions, revising our presentation according to your suggestions, etc., on the revised version of our paper.

---

### Official Review · Reviewer_MsCy · 2024-07-10

**Soundness:** 3
**Presentation:** 3
**Contribution:** 2
**Rating:** 5
**Confidence:** 4

**Summary:**

This paper introduces SFCNet, a spherical frustum sparse convolution network designed for semantic segmentation of LiDAR point clouds. Traditional 2D projection methods suffer from quantized information loss when multiple points project onto the same pixel, leading to sub-optimal segmentation. To address this, the authors propose a spherical frustum structure that retains all points within the same pixel by using a hash-based representation. This structure is then utilized in spherical frustum sparse convolution, which considers both neighboring frustums and the nearest point within each frustum for convolution. Additionally, Frustum Farthest Point Sampling (F2PS) is used to sample points stored in spherical frustums. Experimental results on the SemanticKITTI and nuScenes datasets demonstrate that SFCNet outperforms conventional 2D projection-based methods, particularly in the segmentation of small objects, while maintaining complete point cloud information.

**Strengths:**

1. The paper is well-written and easy to follow. The introduction to the Spherical Frustum Sparse Convolution and Frustum Farthest Point Sampling is clear and comprehensible. It provides a clear narrative structure supported by comprehensive figures and tables. The experimental settings are reasonable and well-documented.
2. The motivation to address quantized information loss during projection is well-founded.

**Weaknesses:**

1. The experimental results do not demonstrate a clear superiority of the proposed method. While the SFCNet shows some mIoU improvements over prior projection methods on the SemanticKITTI and nuScenes datasets, these gains are marginal. Furthermore, the performance of SFCNet is still significantly below that of SOTA 3D voxel-based methods.
2. The paper does not provide an analysis of the computational efficiency of the proposed method. Given that SFCNet involves specific point sampling for convolution, assessing the computational efficiency is crucial to understand its practical significance and potential trade-offs in real-world applications.

**Questions:**

In Table 1, it is clear  that 3D voxel-based methods achieve significantly better results compared to point-based and 2D projection-based methods. Does this suggest that 3D voxel-based methods are inherently superior?

Could you provide a comparison of the computational efficiency of SFCNet against other baseline methods? This information is crucial to evaluate the practical applicability and performance trade-offs of SFCNet.

**Limitations:**

No limitations are discussed in the paper.

---

> ### Author Rebuttal · Authors · 2024-08-07
>
> We sincerely appreciate your valuable suggestions. We are encouraged by your confirmation of the presentation of our paper and our motivation for overcoming quantized information loss. The following are our responses to your concerns:
>
> ---
>
> ### Q1: Performance Gain Compared to Prior 2D Projection-based Methods.
>
> Though the performance gains of the mIoU metric are marginal, SFCNet achieves significant performance improvement on small objects compared to the previous 2D projection-based methods as shown in Tables 1 and 2 in the main manuscript, especially on the human, motorcycle, etc. The improvement mainly results from overcoming quantized information loss by the proposed spherical frustum structure. The better performance of small objects shows the significant practical value of SFCNet for safe autonomous driving.
>
> ---
>
> ### Q2: Performance Gap between SFCNet and 3D Voxel-based Methods.
> As shown in the common response to you in Q2 of Author Rebuttal, SFCNet can realize the trade-off between efficiency and performance and thus has a realistic value in specific scenarios. In addition to the realistic contribution, SFCNet also builds a foundation for future works on the 2D projection-based methods since our work provides the spherical frustum structure to overcome the quantized information loss during 2D projection. At the beginning of deep learning-based point cloud semantic segmentation, the 2D projection-based method is the pioneer that shows reasonable performance for semantic segmentation of outdoor LiDAR point cloud since it can transfer the success of image semantic segmentation to the field of point cloud semantic segmentation. Therefore, though 3D voxel-based methods have better performance than 2D projection-based methods recently, it does not mean the 2D projection-based method is inherently weaker than 3D voxel-based network. If a strong image semantic segmentation backbone is proposed, the proposed spherical frustum structure can be adopted to apply the strong backbone to the field of LiDAR point cloud semantic segmentation without quantized information loss.
>
> ---
>
> ### Q3: Analysis of Computational Efficiency.
> As shown in the common response to you in Q1 of Author Rebuttal, SFCNet shows better computational efficiency against the other baseline methods. Though we have a specific point sampling method, the computational complexity of our point sampling method Frustum Farthest Point Sampling (F2PS) is linear as shown in the L203 in the main manuscript. Thus, both the theoretical and experimental analysis show our method is computationally efficient and shows significant potential for practice application.
>
> ---
> We hope our response can address your concerns about the performance and computational efficiency between our SFCNet and the other baseline methods. If you have further problems, please let us know.

---

### Official Review · Reviewer_bPaa · 2024-07-11

**Soundness:** 3
**Presentation:** 3
**Contribution:** 3
**Rating:** 6
**Confidence:** 4

**Summary:**

The paper introduces a Spherical Frustum Sparse Convolution Network (SFCNet), a novel approach for LiDAR point cloud semantic segmentation that addresses the quantized information loss existing in 2D projection-based methods. By utilizing a spherical frustum structure, the SFCNet preserves all points within a 2D projection area, thus maintaining the complete geometric structure of the data. This model improves the segmentation accuracy, particularly for small objects, through the innovative use of hash-based spherical frustum representation and a custom sampling method called Frustum Farthest Point Sampling (F2PS). Extensive evaluations on the SemanticKITTI and nuScenes datasets demonstrate superior performance over existing methods.

**Strengths:**

S1. Originality of Spherical Frustum Structure. The spherical frustum structure is a novel approach that effectively preserves all the points within a given 2D projection area, mitigating the common issue of quantized information loss.

S2. Efficiency of Representation. The proposed hash-based spherical frustum representation ensures memory efficiency and computational feasibility.

S3. Performance Boost. The enhanced accuracy in segmenting small objects and maintaining geometric integrity significantly contributes to the field, especially for applications in autonomous driving and robotic navigation.

**Weaknesses:**

W1. Performance Decrease on Large Classes. While SFCNet introduces significant advancements in semantic segmentation of LiDAR data, the model demonstrates slightly weaker performance on wide-range classes, such as roads and parking areas. This suggests that while the model excels in handling small object segmentation, its techniques might not be as effective for broader landscapes or larger objects.

**Questions:**

Q1a. Performance Change on Large Classes. Could the authors provide insights into why SFCNet shows slightly weaker performance on wide-range classes, such as roads and parking areas (see also W1)?

Q1b. Maintaining performance on large Classes. Are there any thoughts how the model could improve the small classes while maintaining performance on larger classes?

Q2. Performance Comparison toward tail-end of Distribution. Given a different performance w.r.t. class sizes, additional tests on accuracy w.r.t. data changes might be helpful. How does SFCNet adapt its segmentation accuracy in non-standard environments (like in night settings, heavy rain, etc.)? Insights here could greatly influence its applicability in real-world settings.

**Limitations:**

Investigating how the system performs in unpredictable, non-standad settings would offer a clearer picture of its real-world readiness. The paper can also benefit from a deeper exploration of SFCNet's slightly weaker performance on wide-range classes such as roads and parking areas. Understanding the factors that contribute to these discrepancies could lead to improvements in the model's overall accuracy and applicability in real-world scenarios.

---

> ### Author Rebuttal · Authors · 2024-08-07
>
> We sincerely appreciate your valuable suggestions. We are encouraged by your confirmation of our spherical frustum structure, efficiency, and performance improvement on small objects. The following are our responses to your concerns:
>
> ---
> ### Q1: Insights and Improvement Direction on Performance of Wide-Range Classes.
> - **Discussion on Performance:** To implement the 2D convolution on the spherical frustum, only the nearest points in the neighbor spherical frustums are adopted in the spherical frustum sparse convolution. This design may limit the receptive field of the network and thus result in a slightly weaker performance of the wide-range classes.
> - **Improvement Direction:**  The performance improvement on small classes mainly results from the proposed spherical frustum structure, which overcomes quantized information loss and preserves the complete geometry structure. Thus, to maintain the performance on both the wide-range classes and small classes, the direction is to expand the receptive field based on our spherical frustum structure. To realize this, future work can lie in combining the vision network architecture with a larger receptive field, like the vision transformer [1] or vision mamba [2], with our spherical frustum structure.
>
> ---
> ### Q2: Additional Tests on Accuracy w.r.t Data Changes.
> We adopt two methods to change the data size, including increasing instance class sizes by instance augmentation [3] and decreasing labeled class sizes using the ScribbleKITTI [4] dataset. The number of instances before and after increasing instance class sizes are shown in the first table below. In addition,  the second table below shows how many labeled million points belong to each class in the original and scribble datasets.
> || car    | bicycle | motorcycle | truck | other-vehicle | person | bicyclist | motorcyclist |
> | -| ------ | ------- | ---------- | ----- | ------------- | ------ | --------- | ------------ |
> | Before Increasing | 149531 | 4231    | 2835       | 2543  | 7072          | 7287   | 1390      | 532|
> | After Increasing  | 149575 | 5809    | 4387       | 5846  | 8433          | 8208   | 5437      | 4535         |
>
>
> || car  | bicycle | motorcycle | truck | other-vehicle | person | bicyclist | motorcyclist | road  | parking | sidewalk | other-ground | building | fence | vegetation | trunk | terrain | pole | traffic-sign |
> | -------- | ---- | ------- | ---------- | ----- | ------------- | ------ | --------- | --| ----- | ------- | -------- | ------------ | -------- | ----- | ---------- | ----- | ------- | ---- | ------------ |
> | Original | 99.4 | 0.392   | 0.94       | 4.59  | 5.46          | 0.817  | 0.299     | 0.088        | 467.1 | 34.6    | 338.2    | 9.17         | 311.8    | 170.0 | 627.2      | 14.2  | 183.6   | 6.71 | 1.44         |
> | Scribble | 6.57 | 0.07    | 0.18       | 0.24  | 0.43          | 0.16   | 0.05      | 0.02         | 16.1  | 1.93    | 24.7     | 0.895        | 25.5     | 38.5  | 60.5       | 2.76  | 9.15    | 1.35 | 0.27         |
>
> The experiments are conducted on the two cases. As shown in the table below, increasing the instance class sizes can improve the segmentation performance, since more instances of the rare classes are added to ease the long-tailed problem. As for decreasing the labeled class sizes, the performance also decreases since the supervision is weaker with fewer labels.
>
> | Method| mIoU (%) $\uparrow$ |
> | -| -|
> | SFCNet (Ours)| 62.9                |
> | SFCNet (Ours) w/ increment of instance class sizes | 63.2                |
> | SFCNet (Ours) w/ decrease of whole class sizes     | 56.0                |
>
> ---
> ### Q3: How SFCNet Adapts Accuracy in Non-Standard Environments.
>
> Robo3D [5] dataset provides the LiDAR point cloud adding simulated noises in non-standard environments and corresponding semantic labels on the SemanticKITTI validation set. However, the night and heavy rain environments are not provided by Robo3D. We do not find any other datasets that provide the data in the night and heavy rain environments modified from the SemanticKITTI dataset. Thus, to fairly test how our SFCNet and the baseline model adapt accuracy in the non-standard environments with the same point cloud data, the snow and wet ground environments on the Robo3D dataset are adopted. Actually, the snow and wet ground are similar to the dropping rain and wet ground in the heavy rain environment. Meanwhile, the night environment does not cause noise in the LiDAR point cloud since the LiDAR sensor captures the range information by actively sending the LiDAR ray. As shown in the table below, when tested on noisy non-standard environments, the performance of our SFCNet decreases. However, it still has a better performance than the baseline model due to overcoming the quantized information loss. These results indicate the robustness of our method to overcome quantized information loss in various real-world settings.
>
> | Method        | mIoU (%) $\uparrow$ (Snow) | mIoU (%) $\uparrow$ (Wet Ground) |
> | ------------- | -------------------------- | -------------------------------- |
> | Baseline      | 44.2| 53.9|
> | SFCNet (Ours) | 47.5| 55.8|
>
> ---
> ### Reference
> [1] Z. Liu _et al._, ‘Swin transformer: Hierarchical vision transformer using shifted windows’, ICCV, 2021.
>
> [2] L. Zhu _et al._, ‘Vision mamba: Efficient visual representation learning with bidirectional state space model’, ICML, 2024.
>
> [3] Z. Zhou _et al._, "Panoptic-polarnet: Proposal-free lidar point cloud panoptic segmentation.", CVPR, 2021.
>
> [4] O. Unal _et al._, ‘Scribble-supervised lidar semantic segmentation’, CVPR, 2022.
>
> [5] L. Kong _et al._, ‘Robo3d: Towards robust and reliable 3d perception against corruptions’, ICCV, 2023.
>
> ---
> We hope our response can address your concerns about the performance on non-standard environments, accuracy testing with data changes, and the reasons and improvement direction on the performance of wide-range classes. If you have further problems, please let us know.

---

> > ### Comment · Reviewer_bPaa · 2024-08-09
> >
> > Many thanks for the clarification which addressed all my open questions!
> >
> > I would really encourage the authors to include a discussion along Q1 into the paper to illustrate the limitations of the current approach, too. This may also help to emphasize the Conceptual Novelty in a more dialectic way (c.f. reviewer BSNT).

---

> > > ### Author Response · Authors · 2024-08-09
> > >
> > > Thank you very much for your time and your recognition of our method and response. We promise to include the discussion of Q1 in the final version of our paper. Your valuable suggestions really help us improve the illustration of our conceptual novelty in a more dialectical way.

---

### Official Review · Reviewer_AnTM · 2024-07-12

**Soundness:** 3
**Presentation:** 3
**Contribution:** 3
**Rating:** 7
**Confidence:** 4

**Summary:**

This paper introduces a novel spherical frustum structure for LiDAR point cloud semantic segmentation, addressing the quantized information loss in 2D projection-based methods. By preserving all points within frustums and employing a memory-efficient hash-based representation, the Spherical Frustum sparse Convolution Network (SFCNet) achieves superior segmentation accuracy, particularly for small objects. The proposed Frustum Farthest Point Sampling (F2PS) method ensures uniform sampling and computational efficiency. Extensive experiments on the SemanticKITTI and nuScenes datasets demonstrate SFCNet's improved performance over existing methods.

**Strengths:**

- **Innovation in Data Representation**: The spherical frustum structure preserves all points, avoiding quantized information loss and improving segmentation accuracy, especially for small objects.
- **Memory Efficiency**: The hash-based representation efficiently stores spherical frustums, minimizing memory usage compared to dense grid storage.
- **Improved Sampling Method**: The Frustum Farthest Point Sampling (F2PS) method ensures uniform point cloud sampling, enhancing the retention of key information while maintaining computational efficiency.

**Weaknesses:**

- The proposed methods, including the spherical frustum structure and hash-based representation, add layers of complexity that might pose challenges for practical implementation. Interesting to see the analyze on memory usage, parameter size, and inference time.
- The paper focuses primarily on comparisons with 2D projection-based methods. While it mentions the performance gap with 3D voxel-based methods, a more detailed analysis and direct comparison with state-of-the-art 3D voxel-based approaches would provide a clearer picture.
- The paper lacks a sensitivity analysis of key parameters, such as the stride size in the F2PS, the number of points in the spherical frustums, and the impact of different hash table configurations, etc. Understanding how sensitive the model performance is to these parameters would be valuable for tuning the model for specific applications.
- LiDAR data often contains noise and outliers. The robustness of the proposed method to such imperfections is not thoroughly evaluated. Experiments that introduce varying levels of noise and assess the impact on segmentation performance would be beneficial in demonstrating the method’s robustness.
- The qualitative results, although helpful, are somewhat limited in scope. Providing a more diverse set of visualizations, including different types of scenes (e.g., urban, rural, complex intersections), would offer a clearer picture of the method’s practical performance across various real-world scenarios.

**Questions:**

I overall like the idea of this paper. However, my concerns are focused on the experimental results. If the authors can address my concerns regarding the experiments, I am inclined to raise my score.

**Limitations:**

no potential negative social impact

---

> ### Author Rebuttal · Authors · 2024-08-07
>
> We sincerely appreciate your valuable suggestions for our paper. We are encouraged by your confirmation of our data representation, memory-efficient hash-based representation, and efficient point sampling. The following are our responses to your concerns:
>
> ---
> ### Q1: Analysis of Memory Usage, Parameter Size, and Inference Time.
> Please refer to the common response to you in Q1 of Author Rebuttal.
>
> ---
> ### Q2: Detailed Analysis with the 3D Voxel-based Methods.
> Please refer to the common response to you in Q2 of Author Rebuttal.
>
> ---
> ### Q3: Sensitivity Analysis of Key Parameters.
> We have conducted the experiments for different settings of the stride size in the F2PS, the number of points in the spherical frustums, and the configurations of the hash table.
> - **Stride Size in the F2PS:** Due to the computational resource limitation, the ablation studies of four different settings of the stride sizes in the first downsampling layer's F2PS are conducted, including $(1,2),(2,1),(2,4),(4,2)$. Since the point cloud to be downsampled in the first downsampling layer is the densest, the impact of different stride sizes of the F2PS is the most significant. We promise that a thorough analysis will be conducted in the final version of the paper. The results are shown in the table below. The results show for the first F2PS, the $(2,2)$ stride sizes show a better segmentation performance than the other stride size settings. $(2,2)$ stride sizes suitably downsample the point cloud in the vertical and horizon dimensions. Higher or lower downsampling rates result in the oversampling or undersampling of the point cloud respectively.
>
> 	| Stride Sizes $(S_h,S_w)$ | mIoU (%) $\uparrow$ |
> 	| -| -|
> 	| (2,1) | 60.7|
> 	| (1,2) | 62.4 |
> 	| (2,4) | 62.3  |
> 	| (4,2) | 60.5 |
> 	| (2,2) (Ours SFCNet) | 62.9 |
> - **Number of Points in the Spherical Frustums:**  In the spherical frustum structure, the number of points in the frustum is unlimited and only depends on how many points are projected onto the corresponding 2D location. To analyze the effect of the number of points in the frustum, we set the maximal number of points in each spherical frustum and the points exceeding the maximal point number are dropped. As shown in the table below, preserving more points in the spherical frustum results in better segmentation performance, since more complete geometry information is preserved. These results further indicate the significance of overcoming quantized information loss in the field of LiDAR point cloud semantic segmentation.
>
> 	| Maximal Number of Points in Spherical Frustum | mIoU (%) $\uparrow$ |
> 	| - | -|
> 	| 2 | 61.0|
> 	| 4| 61.9 |
> 	| Unlimited (Ours SFCNet) | 62.9 |
> - **Configuration of the Hash Table:** The number of hash functions is the main parameter of the hash table, which means the number of functions used for the hash table retrieval. In the implementation, if the first hash function can successfully retrieve the location of the target point, the other functions will not be used. We change the number of hash functions to show the model sensitivity of hash table configurations.  As shown in the table below, the performance and inference time of SFCNet have little difference under different numbers of hash functions. The results show that in most cases, the first function can successfully retrieve the location, and thus the inference times change slightly in different function numbers. These results indicate that SFCNet is robust to different hash table configurations.
>
> 	| Number of Hash Functions | Inference Time (ms)   $\downarrow$ | mIoU (%) $\uparrow$ |
>   | - | - | -|
> 	| 2| 59.5| 62.9 |
>   | 3 | 60.1 | 62.9                |
> 	| 5| 59.5 | 62.9                |
> 	| 4 (Ours SFCNet)  | 59.7| 62.9                |
>
> ---
> ### Q4: Performance With Different Levels of Noise in LiDAR Data.
> The gaussian noises with zero mean and different standard deviations, including 0.25m and 0.5m, are added to the original LiDAR point cloud to simulate different levels of noise. As shown in the table below, though the segmentation performance decreases while the level of noise increases, our SFCNet has better performance than the baseline model in all cases. Thus, the proposed SFCNet is more robust to point cloud noises than the baseline model which has quantized information loss.
>
> | Method                       | mIoU (%) $\uparrow$ |
> |-|-|
> | Baseline w/o noise| 59.7 |
> | SFCNet (Ours) w/o noise| 62.9|
> | Baseline w/ 0.25m noise | 43.8|
> | SFCNet (Ours) w/ 0.25m noise | 46.6|
> | Baseline w/ 0.5m noise| 30.7|
> | SFCNet (Ours) w/ 0.5m noise  | 34.8|
>
> ---
> ### Q5: Qualitative Results in More Diverse Types of Scenes.
>  We show the qualitative results in more diverse types of scenes on the SemanticKITTI test set in Fig. 1 of the attached PDF file, including the urban, rural, and complex intersections.  As shown in Fig. 1, compared to CENet [1], SFCNet can better segment the distant persons in the complex intersections scene of Fig. 1(c-1) and urban scene of Fig.1 (a-2), poles in the urban of Fig. 1(a-1) and complex intersections scene of Fig. 1(c-2), and trunks in the rural scenes of Fig. 1(b-1) and (b-2). These results validate the consistently better segmentation performance of SFCNet in various real-world scenarios, especially for 3D small objects.
>
> ---
> ### Reference
> [1] H.-X. Cheng _et al._, ‘Cenet: Toward Concise and Efficient Lidar Semantic Segmentation for Autonomous Driving’, ICME, 2022.
>
> ---
> We hope our response can address your concerns about the experimental results. If you have further problems, please let us know.

---

> > ### Comment · Reviewer_AnTM · 2024-08-12
> > **Raised my rating to ACCEPT**
> >
> > I appreciate the authors' rebuttal and additional experiments. I have read the comments from other reviewers and the related rebuttal. The rebuttal has effectively addressed my main concerns. Additionally, I highly appreciate the responses to Q3 and Q5, and I suggest incorporating them into the supplementary material.
> >
> > Overall, I believe this is a technically solid paper, and I am inclined to accept it - I have raised my rating from borderline accept to **ACCEPT**.

---

> > > ### Author Response · Authors · 2024-08-12
> > >
> > > Thank you very much for your time and your recognition of our method and response. We promise to incorporate our responses to your Q3 and Q5 into the supplementary material in the final version of our paper.

---

### Author Rebuttal · Authors · 2024-08-07

We appreciate all reviewers' efforts and valuable suggestions. We are grateful to receive the positive comments from the reviewers, including memory efficiency (Reviewer AnTM, bPaa, BSNT), innovation in spherical frustum structure (Reviewer AnTM, bPaa), and performance "boost" (Reviewer bPaa, BSNT).

We have responded to the concerns of each reviewer individually. In addition, we also notice the common concerns, including memory and computational efficiency (Reviewer AnTM, MsCy), and performance gap with 3D voxel-based methods (Reviewer AnTM, MsCy), raised by the reviewers. Thus, we provide the common response to these concerns as follows:

---

### Q1:  Analysis of Memory Usage, Parameter Size, and Inference Time (Reviewer AnTM, MsCy).
As shown in the table below, the analysis of the memory usage, parameter size, and inference time are conducted through the comparison between our SFCNet and the other baselines. The normalized memory usage and inference time, computed by dividing the whole memory usage and inference time by the number of thousand points, are also shown.
The results show:

- **Memory Usage:** From the metric of normalized memory usage, the memory efficiency of our SFCNet is better than the point-based methods [1-2] and 3D voxel-based methods [3-4]. As for the 2D projection-based methods, SFCNet has a better memory efficiency than the baseline model and a comparable memory efficiency with RangeViT [5]. The results show that based on memory-efficient hash-based representation, the spherical frustum structure introduces little extra memory consumption and is suitable for practical application.
- **Parameter Size:** Since SFCNet has pure 2D convolutional layers, SFCNet has a much smaller parameter size than the transformer-based methods [4-5] and 3D convolution-based methods [3]. In addition, the comparison between SFCNet and the baseline shows the spherical frustum structure does not introduce extra parameters. As for the comparison with the MLP-based methods [1-2], though the parameter size of SFCNet is larger than the MLP-based methods, SFCNet has better segmentation performance.
- **Inference Time:** Our SFCNet has the smallest normalized inference time among the compared methods. The results show the spherical frustum structure and hash-based representation are also computationally efficient.

| Method           | Points | Memory Usage (M) $\downarrow$ | Normalized Memory Usage (M/K) $\downarrow$ | Parameter Size（M）$\downarrow$ | Inference Time (ms) $\downarrow$ | Normalized Time (ms/K) $\downarrow$ |
| -| -| -| -| -| -| -|
| PointNet++ [1]   | ~45K   | 841                           | 18.69                                      | 3.8  | 131.0 | 2.91                                |
| RandLA [2]       | ~120K  | 1395                          | 11.63                                      | 4.9                             | 212.2                            | 1.77                                |
| Cylinder3D [3]   | ~40K   | 1912                          | 47.80                                      | 214                             | 67.5                             | 1.69                                |
| SphereFormer [4] | ~90K   | 2678                          | 29.76                                      | 124                             | 108.2                            | 1.20                                |
| RangeViT [5]     | ~120K  | 1344                          | 11.20                                      | 91                              | 104.8                            | 0.87                                |
| Baseline         | ~90K   | 1090                          | 12.11                                      | 44                              | 46.4                             | 0.52                                |
| SFCNet (Ours)    | ~120K  | 1386                          | 11.55                                      | 44                              | 59.7                             | 0.49                                |

---

### Q2: Detailed Analysis of Performance Gap with 3D Voxel-Based Methods. (Reviewer AnTM, MsCy).

The 3D voxel-based methods adopt the 3D convolutional kernel to learn the semantic mode in the 3D space. In contrast, SFCNet learns the semantic mode through the 2D convolutional kernel, which has a weaker ability to learn the 3D semantic mode and results in a weaker performance of SFCNet than the 3D voxel-based methods. However, based on the efficiency analysis in Q1, compared to the 3D voxel-based methods, our SFCNet has better computational efficiency, lower memory consumption, and fewer parameters. Thus, though there exists a performance gap between our SFCNet and the 3D voxel-based methods, our SFCNet can realize a trade-off between efficiency and performance and has the potential to be applied in specific scenarios.

---

In addition, we provide an attached PDF, where we show the qualitative results in more diverse scenes in Figure 1 for Reviewer AnTM.

---

### Reference
[1] C. R. Qi _et al._, ‘Pointnet++: Deep hierarchical feature learning on point sets in a metric space’, NeurIPS, 2017.

[2] Q. Hu _et al._, ‘Learning semantic segmentation of large-scale point clouds with random sampling’, TPAMI, 2021.

[3] X. Zhu _et al._, ‘Cylindrical and asymmetrical 3d convolution networks for lidar segmentation’, CVPR, 2021.

[4] X. Lai _et al._, ‘Spherical Transformer for LiDAR-based 3D Recognition’, CVPR, 2023.

[5] A. Ando _et al._, ‘RangeViT: Towards Vision Transformers for 3D Semantic Segmentation in Autonomous Driving’, CVPR, 2023.

---

### Decision · Program_Chairs · 2024-09-25

**Decision:**

Accept (poster)

**Comment:**

After rebuttal and discussion, all four reviewers are positive, but to a varying degree: all scores between borderline accept and strong accept were given. The main strengths are novelty of the representation, the efficiency, and the performance, where the least positive reviewer criticizes the lack of experimental evidence on the latter.